# Pushing the boundaries of optoacoustic microscopy by total impulse response characterization

Markus Seeger [1,2], Dominik Soliman[1,2], Juan Aguirre[1,2], Gael Diot[1,2], Jakob Wierzbowski[3] & Vasilis Ntziachristos [1,2✉]

Optical microscopy improves in resolution and signal-to-noise ratio by correcting for the system's point spread function; a measure of how a point source is resolved, typically determined by imaging nanospheres. Optical-resolution optoacoustic (photoacoustic) microscopy could be similarly corrected, especially to account for the spatially-dependent signal distortions induced by the acoustic detection and the time-resolved and bi-polar nature of optoacoustic signals. Correction algorithms must therefore include the spatial dependence of signals' origins and profiles in time, i.e. the four-dimensional total impulse response (TIR). However, such corrections have been so far impeded by a lack of efficient TIR-characterization methods. We introduce high-quality TIR determination based on spatially-distributed optoacoustic point sources (SOAPs), produced by scanning an optical focus on an axially-translatable 250 nm gold layer. Using a spatially-dependent TIR-correction improves the signal-to-noise ratio by >10 dB and the axial resolution by ~30%. This accomplishment displays a new performance paradigm for optoacoustic microscopy.

[1] Chair of Biological Imaging, TranslaTUM, Technical University of Munich, Ismaninger Straße 22, 81675 Munich, Germany. [2] Institute of Biological and Medical Imaging, Helmholtz Zentrum München, Ingolstädter Landstr. 1, 85764 Neuherberg, Germany. [3] Walter Schottky Institute, Physics Department, Technical University of Munich, Am Coulombwall 4, 85748 Garching, Germany. ✉email: v.ntziachristos@tum.de

In optical microscopy, the point spread function (PSF) describes the effects of diffraction, aberrations, imperfections in the optics, and alterations induced by the signal collection technology on the image captured. In particular, the PSF captures the blurring the system adds to the representation of an object[1–3]. Typically, the PSF is determined by imaging objects much smaller than the system's resolution (e.g., nanospheres) and observing their apparent size. Spatial deconvolution algorithms then exploit the PSF to improve the resolution and the signal-to-noise ratio (SNR) of the images captured. This technique is broadly applied in conventional white-light and fluorescence microscopy, non-linear and Raman-based microscopy, as well as infrared microscopy and optical coherence tomography[1–15]. Optical-resolution optoacoustic (photoacoustic) microscopy is considered to complement the contrast of optical microscopy, because it resolves optical absorption[16–22]. Assessing morphological and functional readings, optoacoustic microscopy is often implemented in hybrid mode with confocal, multiphoton, or light sheet microscopy, to offer additional contrast mechanisms and improve the overall information content captured[16–22]. Optoacoustic microscopy employs focused optical excitation and achieves resolution and penetration depth similar to those of optical microscopy, suitable for visualizing morphological characteristics of light-absorbing moieties, such as microvasculature, cells, or cellular components[23–27]. In particular, light of transient intensity (e.g., laser pulses) is absorbed and excites ultrasound waves through thermoelastic expansion that are captured by ultrasound detectors.

In analogy to optical microscopy, characterizing the optoacoustic PSF could facilitate improvement in the overall imaging performance. However, the optoacoustic PSF differs from the optical PSF in four ways: (1) optoacoustic signals exhibit a time profile carrying depth information, whereas optical signals are time-independent and represent a single spatial point; (2) the optoacoustic PSF is defined laterally by the optical focus and axially by the detector's characteristics, while the optical PSF is determined only by the optical focus; (3) ultrasound detectors induce significantly stronger spatially-dependent aberrations to the recorded signals compared with optical detectors; (4) optoacoustic signals are bipolar (positive and negative contributions), while optical signals have only positive intensities. These differences mean that optoacoustic microscopy signals must be corrected in a 3D spatially-dependent manner and preferably on the raw signals before projection. As the optoacoustic PSF is defined laterally by the optical focus and, therefore, independent of acoustic properties, accurate correction would mainly affect the SNR and the axial projection of the data. The optoacoustic PSF constitutes in this regard the 3D spatial representation of signals from a point source incorporating (1) the optical impulse response describing the characteristics of optical excitation, (2) the spatial impulse response (SIR) capturing the spatially-dependent signal modification by the ultrasound detection, and (3) the spatially-invariant electric impulse response (EIR) embodying the signal digitization[28–30]. The convolution of these contributions is collectively termed the total impulse response (TIR), a 4D array $(x, y, z, t)$ containing time-resolved optoacoustic impulse responses at all accessible positions.

The determination of the optoacoustic 4D-TIR requires point sources distributed throughout the volume of interest. Early attempts involved imaging absorbing microspheres, focusing an optical beam on a layer of ink or absorbing tape of finite thickness[31,32]. However, these attempts only characterized the impulse response at the acoustic focus and did not yield the 4D-TIR of the system, as they did not address the 3D spatial dependence. Mechanical scans of microspheres in 3D have been suggested to determine spatial dependence[31], but are challenging due to sensitivity issues, high frequencies generated, and weakened photostability after repeated exposure to high-intensity light. Scanning an optical focus across absorbing planes of non-negligible thickness (i.e., larger than half the theoretical axial resolution; ~3 μm at signals of max. 120 MHz) would not generate proper point sources. Needle hydrophones or other ultrasound sources have been considered to actively generate acoustic signals[33,34], but the resulting TIR is distorted by the properties of the ultrasound generator itself. The same holds for pulse-echo characterizations using the investigated sensor for both ultrasound generation and detection. An alternative to experimental TIR determination is computational approaches (i.e., numerical simulations)[30], which fail to capture the characteristics of a particular experimental system, including the electrical response or detector specifications. These difficulties with TIR characterization have limited the application of correction algorithms for improving 3D optoacoustic microscopy. As a result, optoacoustic microscopy images are commonly rendered as frequency-filtered maximum amplitude projections (MAPs), typically displayed as top-view 2D images concealing complex 3D features.

We hypothesized that we could significantly improve the imaging performance of an optical-resolution optoacoustic microscope by applying signal corrections based on the accurately characterized 4D-TIR. For that, we devise a universal method that allows high-quality TIR measurements by distributing true optoacoustic point sources throughout a 3D volume in a highly controlled fashion. To do this, we focused an optical beam onto an ultra-thin absorbing gold layer (GL) to generate spatially-distributed optoacoustic point sources (SOAPs) that are defined laterally by the diffraction-limited optical excitation and axially by the light penetration into the absorbing layer (i.e., ~40 nm at optical excitations of 532 nm). We choose a layer thickness of 250 nm to be thin enough for weak signals to transmit through and thick enough to prevent ablation by permanent optical excitation. Although the GL can be treated as an infinite plane, the optical focus excites a volume so small that SOAPs emit optoacoustic signals analogous to real point sources. By scanning the optical excitation laterally and the acoustic detector axially, we capture the entire 4D-TIR of the microscope. The 4D-TIR characterization based on SOAPs enables the first spatially-dependent signal correction of optoacoustic microscopy, yielding unprecedented improvements, including superior performance in SNR and axial resolution. We used a spatial matched filter (SMF) that cross-correlates the raw optoacoustic signals with the spatially-associated impulse responses as contained in the 4D-TIR. This new performance resolves in vivo and in vitro specimens with a depth discrimination ability not previously available to optoacoustic microscopy. We discuss how SOAP-enabled TIR correction brings new abilities in the performance of optoacoustic microscopy and thereby opens up new application opportunities for basic, translational, and clinical research.

## Results

**Spatially distributed optoacoustic point sources (SOAPs).** We illuminated a 250-nm thick GL by scanning a diffraction-limited focus at a wavelength of 532 nm and a pulse duration of 1.4 ns to generate SOAPs as schematically depicted in Fig. 1a. The numerical aperture of the equipped objective lens of 0.45 led to an illumination spot on the GL with a diameter of ~600 nm. At the laser wavelength used, the light penetrated ~40 nm into the GL (also known as the skin depth), defining the axial extent of the excited volume. The optical focus and, thus, the generated SOAPs were raster scanned throughout a 2D plane of $630 \times 630\ \mu m^2$ using galvanometric mirrors, covering the entire sensitivity field of the 50-MHz ultrasound detector employed (Fig. 1b). For

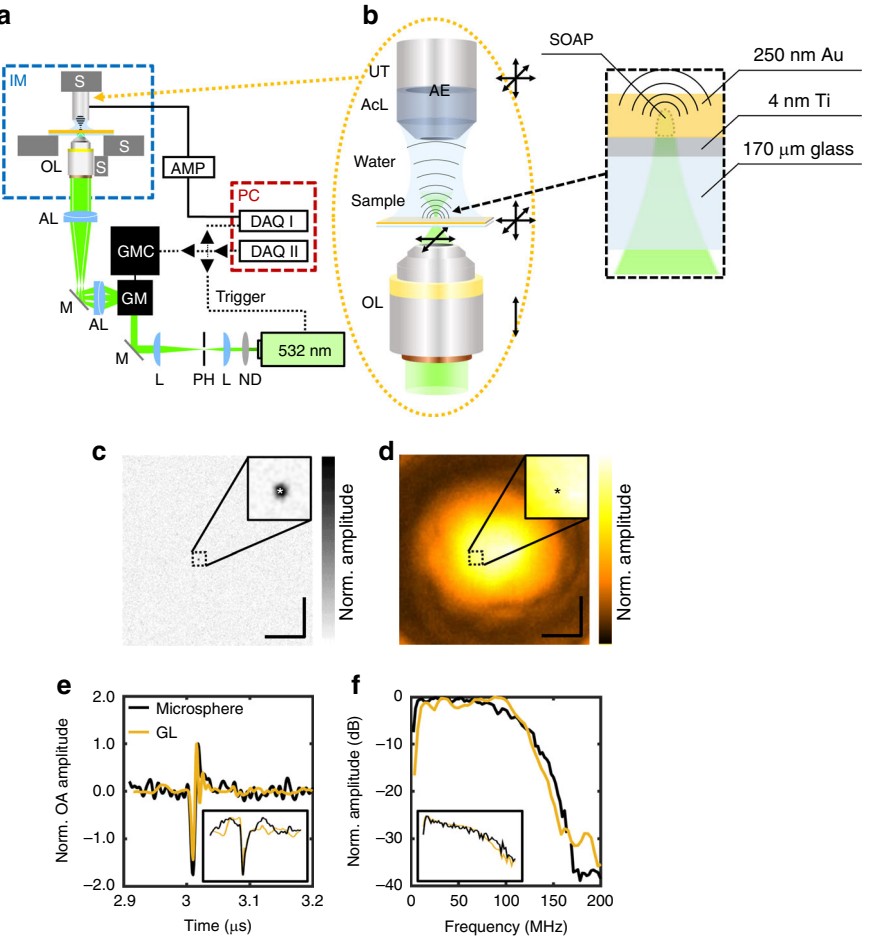

**Fig. 1 The OR-OAM system used for TIR characterization utilizing signals from a gold layer and a 1-μm black polystyrene microsphere. a** OR-OAM system based on raster scanning an optically focused 532-nm laser excitation across a sample placed in an inverted microscope using galvanometric mirrors. **b** Close-up depiction of the arrangement of the imaging framework and schematic depiction of the generation of SOAPs; arrows indicate degrees of freedom within the OR-OAM system. **c** OR-OAM MAP onto the *xy*-plane of a black polystyrene microsphere. **d** OR-OAM MAP of a gold layer at the same axial position. **e** Raw OR-OAM signals and corresponding impulse responses (inset) of the positions in (**c**) and (**d**) marked with asterisks. **f** Frequency content of the microsphere (black) and the gold layer (gold), and their corresponding frequency responses (inset). Scale bar, 100 μm. AcL acoustic lens, AE active element, AL achromatic doublet lens, AMP low noise amplifier, DAQ data acquisition card, GM galvanometric mirror scanner, GMC GM controller, IM inverted microscope, L plano-convex lens, M dielectric mirror, ND neutral density filter, OL microscope objective lens, PH pinhole, S high-precision motorized stage, SOAP spatially distributed optoacoustic point source, UT ultrasound transducer.

validation of the signals obtained from SOAPs, we compared the measurements from the GL scanning with measurements from a 1-μm black polystyrene microsphere embedded in agar as a real optoacoustic point source. In both cases, we applied the identical acquisition parameters (pulse energy, lateral and axial position, number of averages, sampling rate, frequency filtering, and projection). The MAP of the microsphere (Fig. 1c) contained the microsphere as a single optoacoustic point source, while the MAP of the GL scan (Fig. 1d) was built of adjacent SOAPs and, thus, constituted optoacoustic point sources at each pixel. Scanning the GL at a given axial distance to the transducer consequently characterized several orders of magnitude more spatial locations of the system's TIR than an equivalent scan of a single microsphere. Hence, determining the entire TIR in 3D only requires axial displacement of the GL in combination with 2D lateral scans, which constitutes a significantly faster process than a mechanical scan of the microsphere. Furthermore, Fig. 1d illustrates, that a 2D scan of the assumingly homogenous GL appears inhomogeneous, which could indicate imperfections in the transducer.

Good agreement was observed between the microsphere and the corresponding position at the GL in terms of the raw time-resolved A-scan (Fig. 1e) and its frequency content (Fig. 1f). However, much higher SNR was obtained with the GL (28.3 dB) compared with the microsphere measurements (20.7 dB) via comparing the maximum amplitude of the signal recorded in the time window from 2.9 to 3.2 μs after the trigger signal to an identical recording with a blocked laser beam. The microsphere and GL were associated with a similar impulse response, yielded via time-wise integration[30], (Fig. 1e, inset) and frequency response as the impulse response's Fourier transform (Fig. 1f, inset), which are still convolved with the electric impulse and electric frequency response, respectively. We analyzed the similarity between the signals recorded using the GL and the microsphere as possible point sources via the relative squared $L^2$-error defined as $\frac{\left\|Sig_{GL} - Sig_{bead}\right\|_2^2}{\left\|Sig_{bead}\right\|_2^2}$ as it is invariant with respect to Fourier transform (Parseval's identity), which is relevant for the contained frequency content. Such analysis yielded a relative error of 0.28, whereas analogous analysis between the

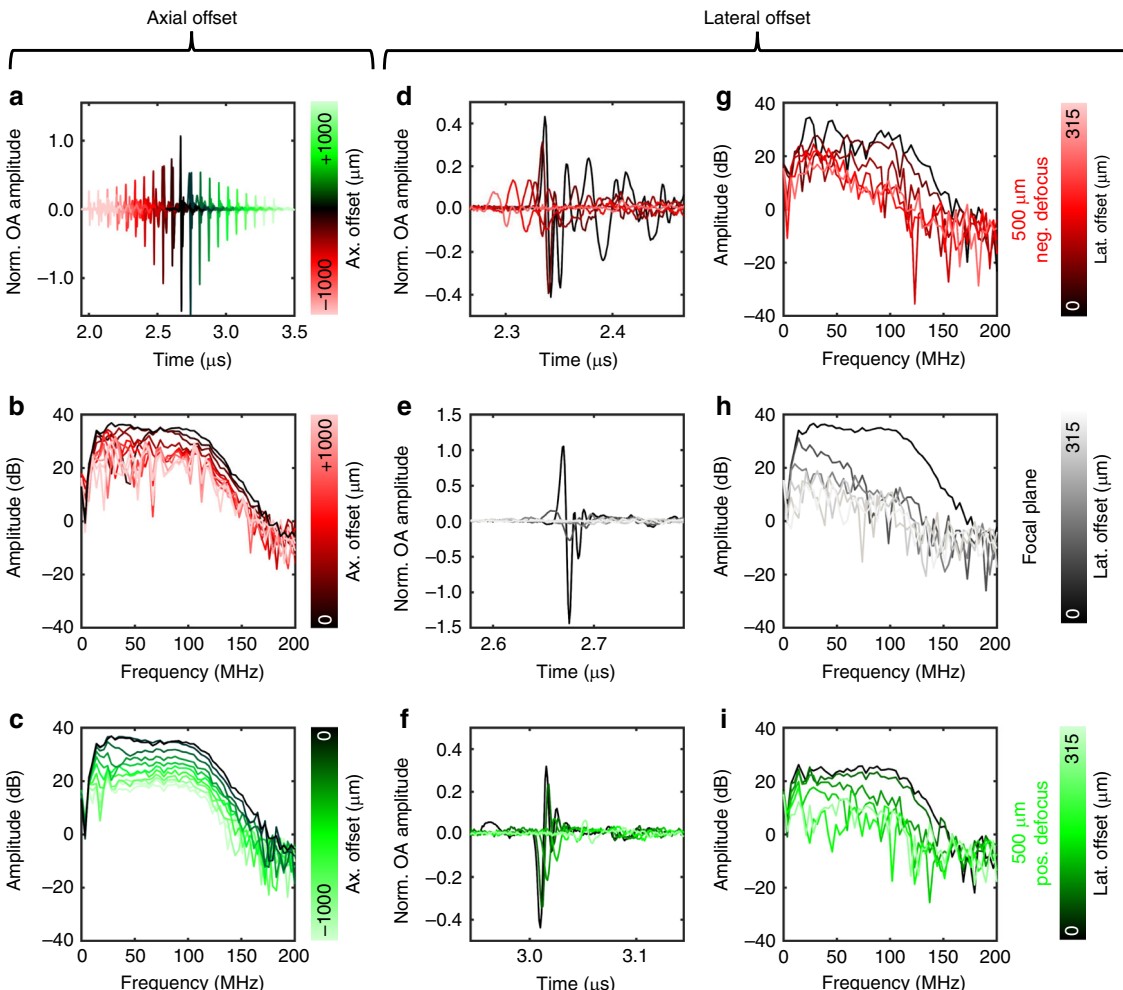

**Fig. 2 Dependence of raw OA A-lines and frequency contents on axial and lateral offset. a** Raw OA A-lines of spatially-distributed OA point sources generated coaxially to the transducer from −1000 μm (red) to the focal plane (black) and +1000 μm (green) defocus in 100 μm steps. Frequency contents of coaxial OA point sources in (**a**) in the direction of **b** negative or **c** positive defocus. Lateral offset dependence of raw OA A-lines from 0 to 315 μm measured at **d** 500-μm negative defocus, **e** the focal plane, and **f** 500-μm positive defocus. **g–i** Frequency contents of the A-lines shown in (**d–f**). ax. axial, lat. lateral, neg. negative, norm. normalized, pos. positive.

microsphere and a nonpoint source specimen (slab) yielded 0.90. These results suggested that by axially translating the transducer with respect to the GL, we could generate SOAPs throughout the volume of interest, which would allow us to characterize the TIR throughout the entire sensitivity field of the transducer.

**Experimental TIR characterization.** Using the GL, we proceeded to characterize in detail the TIR of our optoacoustic microscopy system for later correction of the optoacoustic microscopy images. First, we characterized spatial dependencies of the impulse and frequency responses. We translated the transducer stepwise along the axial direction toward negative offset (i.e., closer to the GL; acoustic near field) and toward positive offset (i.e., farther from the GL; acoustic far field). At each axial position, we analyzed the signals gradually from the coaxial location (i.e., centered beneath the transducer) to lateral offset (away from the central axis). The optical focal plane and GL position were kept constant throughout this procedure, allowing identical SOAPs to be generated at all positions analyzed.

Figure 2 depicts the dependence of raw optoacoustic A-lines as well as their corresponding frequency contents on both axial and lateral offsets with respect to the acoustic focus. When the SOAPs were coaxial (Fig. 2a), a double-sided parabolic optoacoustic amplitude profile was observed along the axial direction, which peaked at the focal plane (black line). In the case of negative defocus (Fig. 2b) or positive defocus (Fig. 2c), the frequency contents deviated from the detectable bandwidth and sensitivity at the acoustic focus (black lines). In the negative defocus region, significant spatially-dependent variation in frequency sensitivity was observed; in contrast, the frequency profile was similar between the positive defocus area and the acoustic focus, but the overall frequency sensitivity was lower in the positive defocus area.

After analysis of axial offset dependence, we examined the dependence of the optoacoustic signal on lateral offsets of 0–315 μm from the central axis of the transducer. At 500 μm negative defocus (Fig. 2d), raw optoacoustic A-lines revealed signal repetitions; at the focal plane (Fig. 2e), the A-lines showed a typical N-shape; and at 500 μm positive defocus (Fig. 2f), the A-line polarity reversed such that the first peak was negative. A-lines in the negative defocus area were more affected by repetitions than A-lines in the positive defocus area. Similarly, the frequency contents in the negative defocus area (Fig. 2g) showed multiple disordered frequency sensitivity dips, whereas the contents in the

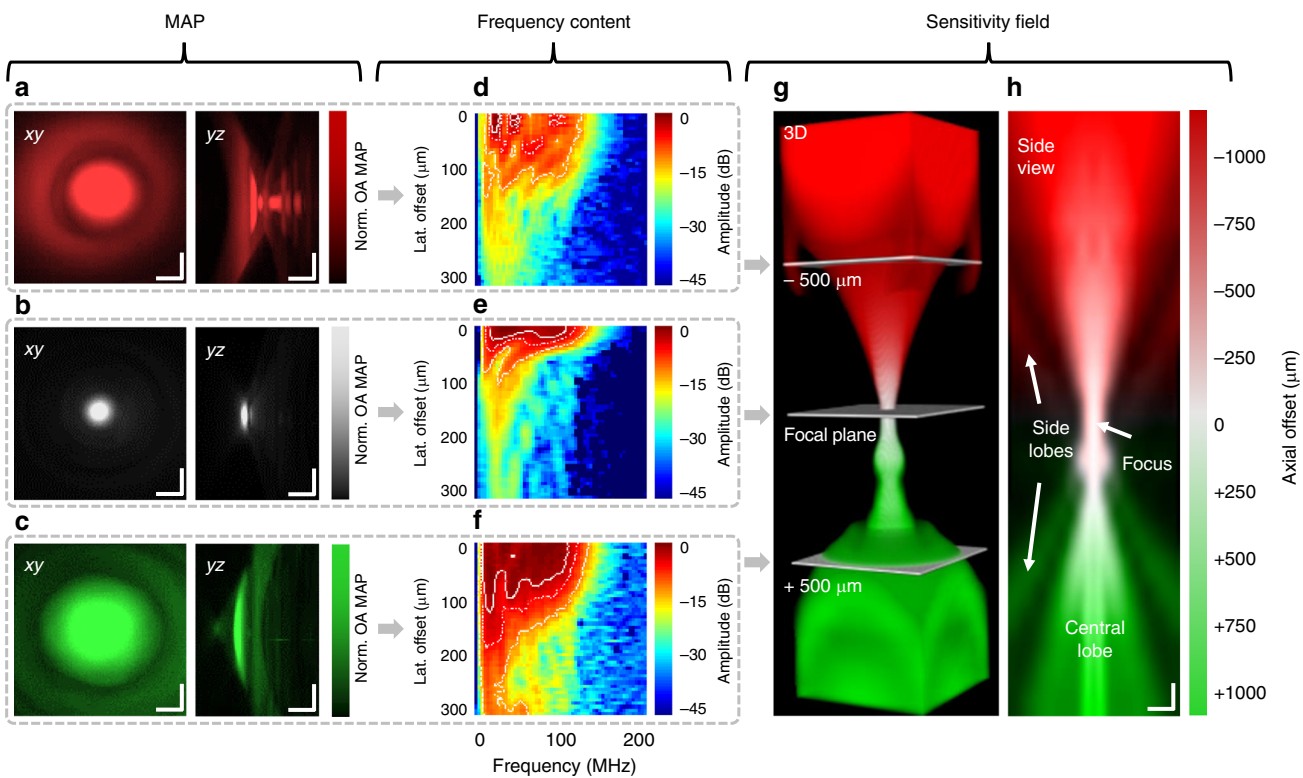

**Fig. 3 Experimental determination of the TIR of a spherically focused 50-MHz transducer.** The TIR was determined by raster scanning the optical excitation across a GL. Raw OR-optoacoustic microscopy MAP onto the *xy*- (left panel) and *yz*-plane (right panel) when the transducer was **a** 500 μm negatively defocused (red), **b** in-focus (white), and **c** 500 μm positively defocused (green). **d–f** Frequency contents for the three cases. Dashed lines show boundaries of −3, −6, and −12 dB. **g** Joined 3D sensitivity field of the transducer. **h** Side view of the central plane of the 3D sensitivity field. Scale bars, 100 μm. lat. lateral, MAP maximum amplitude projection, norm. normalized, OA optoacoustic.

positive defocus area (Fig. 2i) showed more homogeneous frequency capture, similar to the contents at the acoustic focus (Fig. 2h).

Given these capabilities of our GL approach, we set out to provide the first complete determination of the 4D-TIR of a high-frequency ultrasound transducer based entirely on experimental measurements of SOAPs. Figure 3 shows representative optoacoustic signal amplitudes and frequency contents generated by GL-based SOAPs in 2D planes located 500 μm above the acoustic focal plane in negative defocus (red), at the focal plane (white), and 500 μm below the focal plane in positive defocus (green). In these experiments, MAPs in the *xy*- and *yz*-planes were calculated after frequency filtering and Hilbert transformation of the optoacoustic signal. The MAPs in negative defocus (Fig. 3a) showed spatial side lobes of the sensitivity field, visible as concentric rings around the central lobe, as well as multiple temporal-axial artifacts, which lead to image aberrations during reconstruction. The spatial side lobes and temporal artifacts were not observed in MAPs at the acoustic focal plane (Fig. 3b), for which measurements of the full width at half maximum (FWHM) indicated an acoustic resolution of the transducer of ~35 μm laterally and ~6.3 μm axially. The *xy*-MAP in positive defocus (Fig. 3c) showed a spatial side lobe in the *xy*-plane and a curved surface and multiple reflections in the *yz*-plane.

To complement the MAP analysis, we plotted the frequency content as 2D maps for the three cases of defocus or focus. At negative defocus (Fig. 3d), a distinct pattern in the captured frequency content was observed, which varied with lateral offset. At the focal plane (Fig. 3e), frequencies were captured

homogeneously in the central region, and they showed a −6-dB boundary from 10.7 to 119.5 MHz, revealing the actual bandwidth of the transducer at the acoustic focus. At positive defocus (Fig. 3f), frequencies showed a less pronounced pattern than at negative defocus, and high frequencies were lost as the lateral offset increased. These TIR experiments with the GL detected higher frequencies of optoacoustic signals as well as a broader bandwidth for the transducer than the manufacturer's pulse-echo characterization (see Supplementary Fig. 9). This may reflect on one hand the fact that ultrasound waves passed only once through the focal length between transducer and sample in the GL approach but twice in pulse-echo characterization. On the other hand, the characteristics of the transducer of converting electric and acoustic signals to each other influence the recordings ones in the GL approach and twice in pulse-echo characterization.

Finally, we determined the total sensitivity field of the transducer by axially joining and interpolating *xy*-MAPs, each individually normalized, over a volume of 630 × 630 × 2300 μm³ (Fig. 3g). The field showed an asymmetric shape with side lobes corresponding to those seen in the individual defocused MAPs. The field also included a knob-like bulge at positive defocus of ~150 μm. This bulge may be explained by the overlap of the spatial side lobes also visible in the side view of the central plane (Fig. 3h). Generating the sensitivity field for certain frequency bands revealed a strong spatial dependence of the frequency coverage (see Supplementary Fig. 6). The spatial side lobes were mostly carrying low frequencies, the bulge seemed to be induced by middle frequencies, and the acoustic focus was dominated by high frequencies.

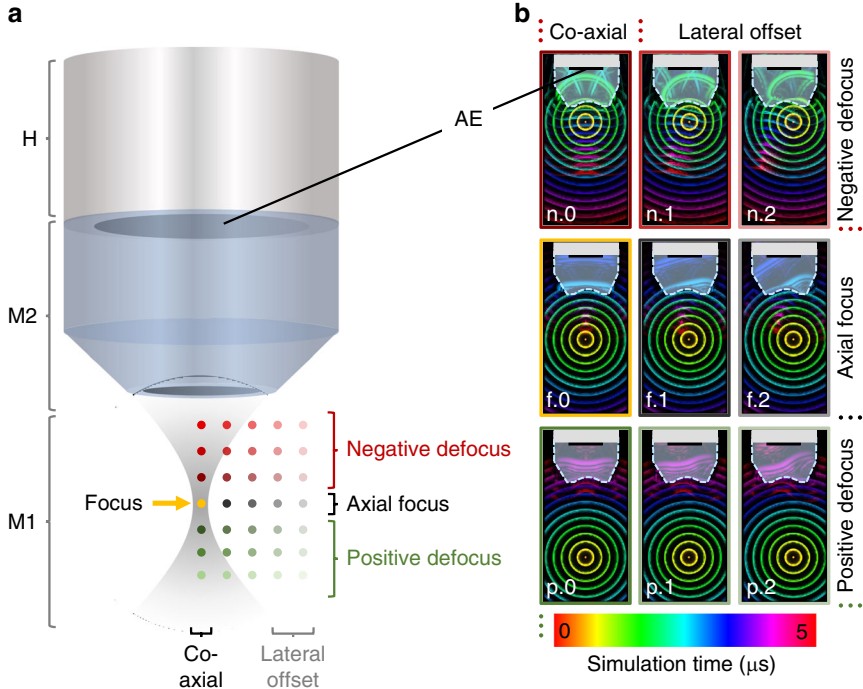

**Fig. 4 Simulation scheme of the TIR for a spherically focused 50-MHz transducer. a** The simulation grid of acoustic point sources consists of two materials M1 and M2 representing the actual settings and geometry of OR-optoacoustic microscopy utilizing a water-coupled spherically-focused transducer by means of a conically shaped plano-concave glass lens attached to the sensor. AE single active element, M1 medium 1 (water), M2 medium 2 (glass), H housing. **b** Representative visualization of TIR color-coded simulation time points for arrangements in axial focus (f.0–2), negative defocus (n.0–2), or positive defocus (p.0–2) at coaxial (*.0) and with lateral offsets (*.1–2). Only in the coaxial and axially focused case (f.0) does a plane wave reach the active element, whereas in axial and lateral offsets (all other panels), interference and wavefront distortion patterns can be observed.

**Numerical TIR simulation**. In order to begin explaining the relevance of the features of the TIR that we observed experimentally, we simulated the 4D-TIR of a 50-MHz transducer using the k-wave package in Matlab[35]. Figure 4a shows schematically the discrete simulation grid with the same geometry as the experimental setup used herein. During the simulation, the initial pressure source was stepwise rastered through a 3D volume identical to the experimental arrangement. The simulation indicated that when the initial pressure source was located at the axial and lateral focus (Fig. 4b, f.0), a plane wave in the glass lens reached the flat active element constituting the actual ultrasound sensing component, hitting the entire element simultaneously. When the pressure source was at a laterally offset within the axial focal plane (Fig. 4b, f.1 and f.2), curved acoustic waves and interferences occurred, leading to different parts of the active element being reached by the waves after than other parts. This may be because the acoustic wavefront arrived at different times at the surface of the concave glass lens and at the inclined rim. At negative defocus (Fig. 4b, n.0–n.2), wavefronts were strongly curved because of the nonplanar shapes of the waves arriving at the glass lens surface, inducing numerous interferences and reflections. The same but opposite bending of wavefronts occurred at positive defocus (Fig. 4b, p.0–p.2); this is because the wavefronts arrived at the outer parts of the glass lens before they arrived at the central part. We further performed a 3D simulation at six selected positions within the simulated volume that includes a 170-μm glass substrate to test the influence of the layered structure of the GL (see Supplementary Fig. 7). Despite small additional signal artifacts that might be induced by acoustic reflections within the glass layer, no significant difference was observed. This simulation confirms the initial assumption of the GL serving as an emanation basis for appropriate OA point sources.

Using the simulation results, we calculated MAPs, 2D frequency content plots, and the 3D sensitivity field (Fig. 5) in the same way that we calculated these results from the experimental TIR characterization in Fig. 3. We found agreement between simulation and experiment to be fairly good overall (see Supplementary Movie 1): the simulated MAPs in negative defocus (Fig. 5a), at the focal plane (Fig. 5b), and in positive defocus (Fig. 5c) resembled the corresponding experimental results in size and appearance of spatial and temporal artifacts (Fig. 3a–c). The MAPs at the focal plane showed the simulated acoustic focus with dimensions of $42 \times 13$ μm. The corresponding frequency contents for negative defocus (Fig. 5d), at the focal plane (Fig. 5e), and positive defocus (Fig. 5f) showed interference patterns analogous to the experimental results (Fig. 3d–f). The frequencies were covered homogeneously at the focal plane (Fig. 5e), with a −6-dB bandwidth from 14.0 to 85.6 MHz. The joined and interpolated MAPs over the full 3D volume (Fig. 5g) as well as the side view of the central plane (Fig. 5h) showed an asymmetric sensitivity field along the central axis.

This convergence of simulation and experiment suggests that observed optoacoustic signal alterations and distortions can be attributed to internal reflections and interferences in the glass lens due to axial and lateral offsets of the signal's origins. However, simulated and experimental TIR diverged in the prominence and shape of spatial and frequency artifacts. In addition, the knob-like bulge was not observed in the simulated sensitivity field. These differences highlight the importance of experimental TIR characterization using the same setup as for the imaging itself. In addition, simulation-based TIR characterization neglects the electrical properties of the data acquisition chain (i.e., the EIR)

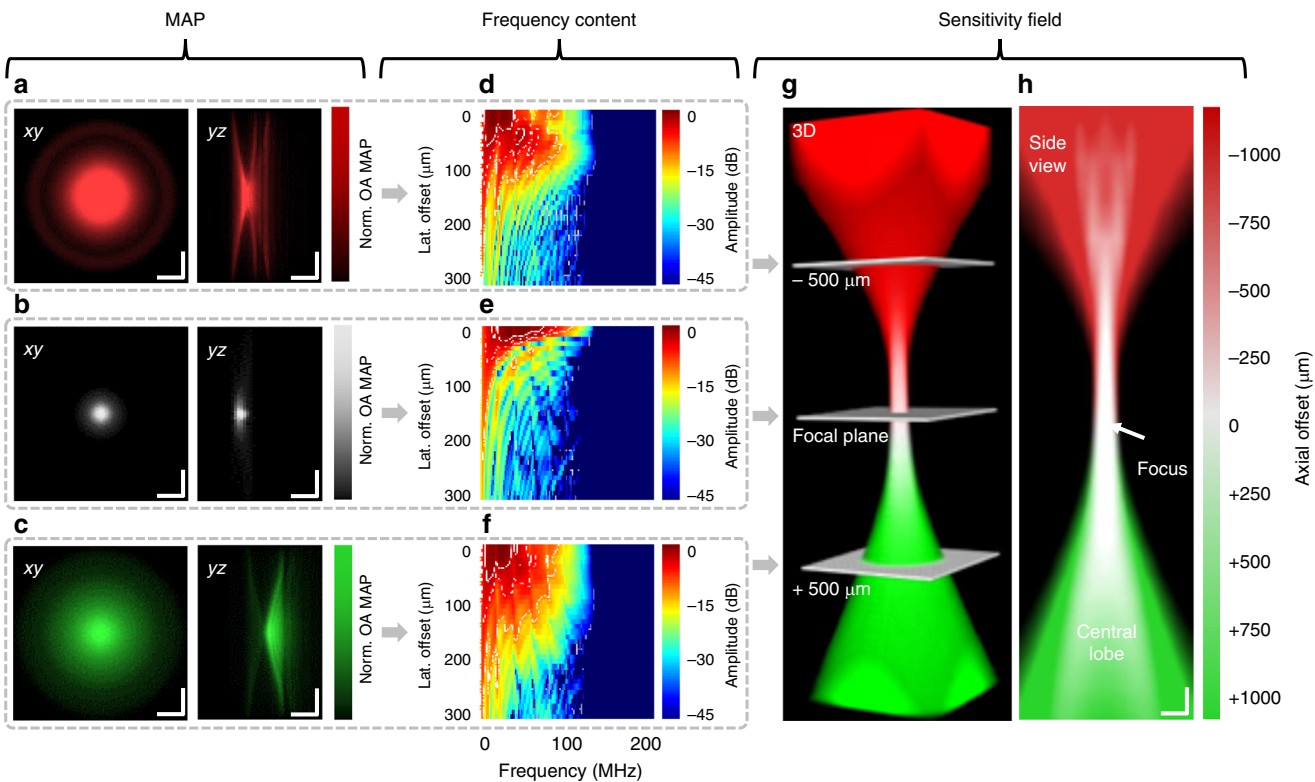

**Fig. 5 Simulation of the TIR of a spherically focused 50-MHz transducer.** TIR determination by simulating point sources throughout a 3D volume. Simulated OR-optoacoustic microscopy MAP onto the *xy*- (left panel) and *yz*-plane (right panel) for an arrangement **a** in 500 μm negative defocus (red), **b** in-focus (white), or **c** in 500 μm positive defocus (green). **d–f** Corresponding frequency contents. Dashed lines indicate boundaries of −3, −6, or −12 dB. **g** Joined 3D sensitivity field of the transducer. **h** Side view of the central plane of the 3D sensitivity field. Scale bars, 100 μm. lat. lateral, MAP maximum amplitude projection, norm. normalized, OA optoacoustic.

and, thus, cannot provide the basis for correcting for such properties or additional convolutions of the signals in optoacoustic imaging.

**TIR correction of in vivo and in vitro imaging.** The detailed characterization of the 4D-TIR described here for the first time for optical-resolution optoacoustic microscopy may be useful for improving theory and device engineering, and even more importantly, it may help to improve the quality of reconstructed images. Therefore, we examined whether our experimental TIR analysis based on SOAPs could be exploited to correct for spatial variations in the optoacoustic signal detection. To this end, we developed a TIR-correction algorithm based on SMF[36–38]. Since SMF methods are linear, they can correct time-of-flight (ToF) differences for signals emanating from locations with lateral offsets, and they can take into account spatial variations in impulse and frequency responses. SMF methods cross-correlate optoacoustic signals with their spatially-corresponding impulse response, transforming their typical bipolar shape into a main positive correlation peak. We compared the quality of the optoacoustic images of a phantom sample consisting of two intertwined 18-μm blacked polystyrene sutures when they were processed using a standard sequence of frequency filtering and Hilbert transformation (Fig. 6a) to a novel sequence of frequency filtering and TIR correction using SMF (Fig. 6b).

In the conventional approach (Fig. 6a), optoacoustic data were filtered according to the frequency bandwidth of the employed transducer, followed by Hilbert transformation, and then back-projection in 3D using the speed of sound in the sample. Repetitive signal shapes were observed in the raw optoacoustic data, the averaged and filtered data, as well as Hilbert-transformed data. The corresponding 3D depiction appeared bent due to the ToF differences when lateral offsets were present, i.e., in the outer parts of the FOV. The 3D depiction also contained reflection artifacts occurring after the main signals, which therefore appeared at lower positions and in different colors in the depth color-coded 3D projection. In contrast, in the TIR-SMF approach (Fig. 6b), filtered and averaged optoacoustic data were cross-correlated with the corresponding signals of SOAPs at the identical spatial positions[39]. After cross-correlation, A-scans were superimposed in time and space. Signal amplitudes were higher than in the conventional approach. Next, negative correlation values were removed by thresholding, and the two correlation side lobes before and after the main lobe were cut out by windowing. Finally, in the 3D depiction, the TIR-SMF approach gave a faithful representation of the phantom as a two circular and homogenously colored sutures.

To test the ability of the TIR-SMF approach to improve optoacoustic microscopy of complex samples, we imaged various reference and unlabeled biological specimens (see Supplementary Figs. 1–4), including the GL, 18 μm blacked polystyrene sutures, a mouse ear vasculature in vivo, an isolated red blood cell (RBC) in vitro, and a zebrafish brain vasculature in vivo. Comparison of optoacoustic microscopy of the GL using the conventional approach (Fig. 7a) and the TIR-SMF approach (Fig. 7b) indicated that applying the TIR-SMF correction reduced field curvature and reflection artifacts, leading to a flat, thin representation of the GL. In the example shown, the GL

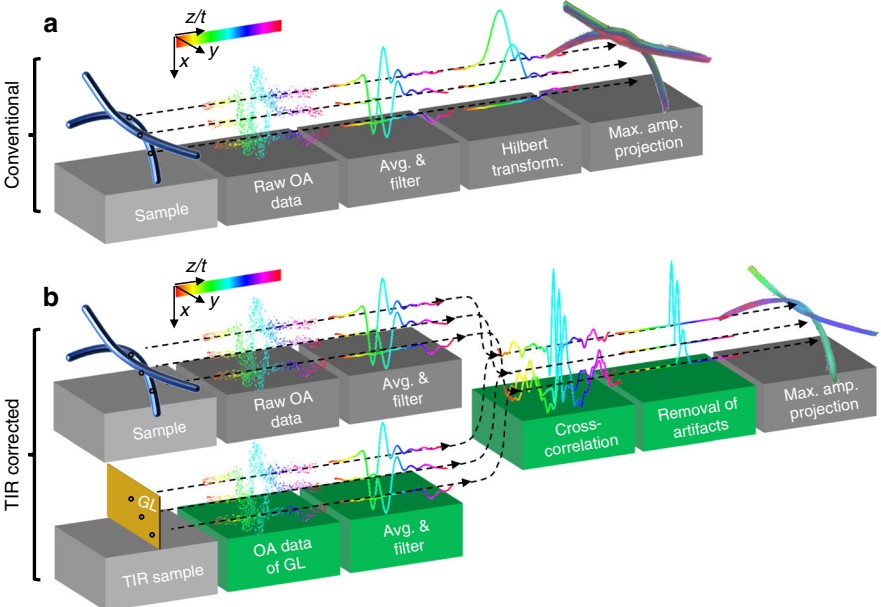

**Fig. 6 Sequence for TIR correction of optoacoustic signals using an SMF.** Scheme of data processing steps of the **a** conventional sequence consisting of signal collection, averaging, frequency filtering, Hilbert transformation, and projection; and **b** the herein developed TIR-corrected sequence, in which the recorded, averaged, and filtered signals are cross-correlated with the signals of SOAPs of same spatial position followed by artifact removal and projection.

served as both the investigated sample and the sample for TIR characterization in two measurements conducted separately. For the conventional approach, the axial thickness of the GL determined by measuring the FWHM was 6.85 µm at a centered position in the FOV and 10.69 µm at a lateral offset of 200 µm, and for the TIR-SMF correction 5.32 and 7.37 µm, respectively. Hence, TIR-SMF corrections achieved an axial resolution improvement of 22.4% at the center and 31.1% at the outer regions of the FOV. The field curvature estimated between the center and a 200-µm lateral offset was 7.1 µm for the conventional approach and <1.5 µm for the TIR-SMF approach. Similarly, comparison of uncorrected (Fig. 7c) and TIR-SMF-corrected (Fig. 7d) optoacoustic microscopy of mouse ear vasculature in vivo showed that TIR-SMF correction eliminated bending of the vasculature and multiple appearances of each vessel along the z-axis (see Supplementary Movie 2). This improvement was also observed for a zoomed-in region of a selected microcapillary (Fig. 7c, d and Supplementary Movie 3). To compare the SNR obtained with or without TIR-SMF correction, we applied the following equation describing the image-SNR improvement ΔSNR:

$$\Delta \mathrm{SNR} = 10 \cdot \log_{10} \left[ \frac{\sum_{x=0}^{n_x-1} \sum_{y=0}^{n_y-1} \left[ a_r(x,y) \right]^2}{\sum_{x=0}^{n_x-1} \sum_{y=0}^{n_y-1} \left[ a_r(x,y) - a_c(x,y) \right]^2} \right], \quad (1)$$

where $a_r$ and $a_c$ denote the amplitude values for the raw and corrected images, respectively, at the pixel positions $x$ and $y$ for all pixels $n_x$ and $n_y$[40]. SNR improvement as a result of the TIR-SMF algorithm was 10 dB for the xy-MAP, 7.2 dB for the xz-MAP, and 5.1 dB for the yz-MAP. The correction also led to more faithful imaging of presumably circular microvessels: conventional reconstruction gave an ellipsoid ratio of 1:1.81 (lateral diameter, 4.7 µm; axial diameter, 8.5 µm), whereas the TIR-corrected reconstruction gave an ellipsoid ratio of 1:1.32 (lateral diameter, 4.6 µm; axial diameter, 6.1 µm). Assuming a circular microcapillary, these results correspond to an axial resolution improvement of 27.1%. A similar comparison between conventional reconstruction and TIR correction was performed for an isolated RBC in vitro (Fig. 7e, f), which was

imaged at the acoustic focus. Whereas the conventional reconstruction revealed an axially elongated RBC, TIR correction achieved a much better representation of an RBC in form of a donut-like shape (see Supplementary Movie 4). Furthermore, the increase in SNR led to a smoother surface boundary. Finally, an unlabeled zebrafish brain vasculature in vivo (Fig. 7g, h) was imaged and similarly compared. TIR correction achieved much sharper vessels and a more realistic geometry of the overall brain vasculature (see Supplementary Movie 5).

The TIR-SMF approach was further compared with alternative methods that correct with the focal impulse response, with the simulated TIR, or with a TIR measured using a slab of absorbing material (with or without ToF correction) (see Supplementary Figs. 1–4). For reference samples (i.e., GL and 18-µm suture) as well as biological specimens (zebrafish brain vasculature and RBC), the TIR-SMF approach led to the most accurate depiction in 3D and the highest SNR. In addition, the TIR-SMF approach was compared with deconvolution approaches both on the signal level and on the image level (see Supplementary Fig. 5). For the former, standard numerical filtering methods like Fourier deconvolution proved to be not flexible enough to deal with the given situation and produced extremely noisy images. Hence, we formulated the deconvolution as a regularized linear least squares problem $f^* := \arg \min_f \| f * \mathrm{IR} - s \|_2^2 + \lambda \| f \|_2^2$, where the regularization parameter was chosen visually from a broad range of values. The optimization was numerically solved with 5000 LSQR iterations to ensure convergence. For the image level, we characterized the systems spatially-dependent PSF using a 1-µm microsphere at seven positions of lateral offset (i.e., 0–300 µm) and used a parallel iterative deconvolution in 3D to apply a modified residual norm steepest descent algorithm with five iterations and spatially-dependent 3D PSFs. The TIR-SMF approach was again found to be superior to these alternatives in spatial representation of the data, noise reduction, and axial resolution improvement.

## Discussion

In this work, we demonstrate TIR-corrected optical-resolution optoacoustic microscopy with markedly superior SNR and axial

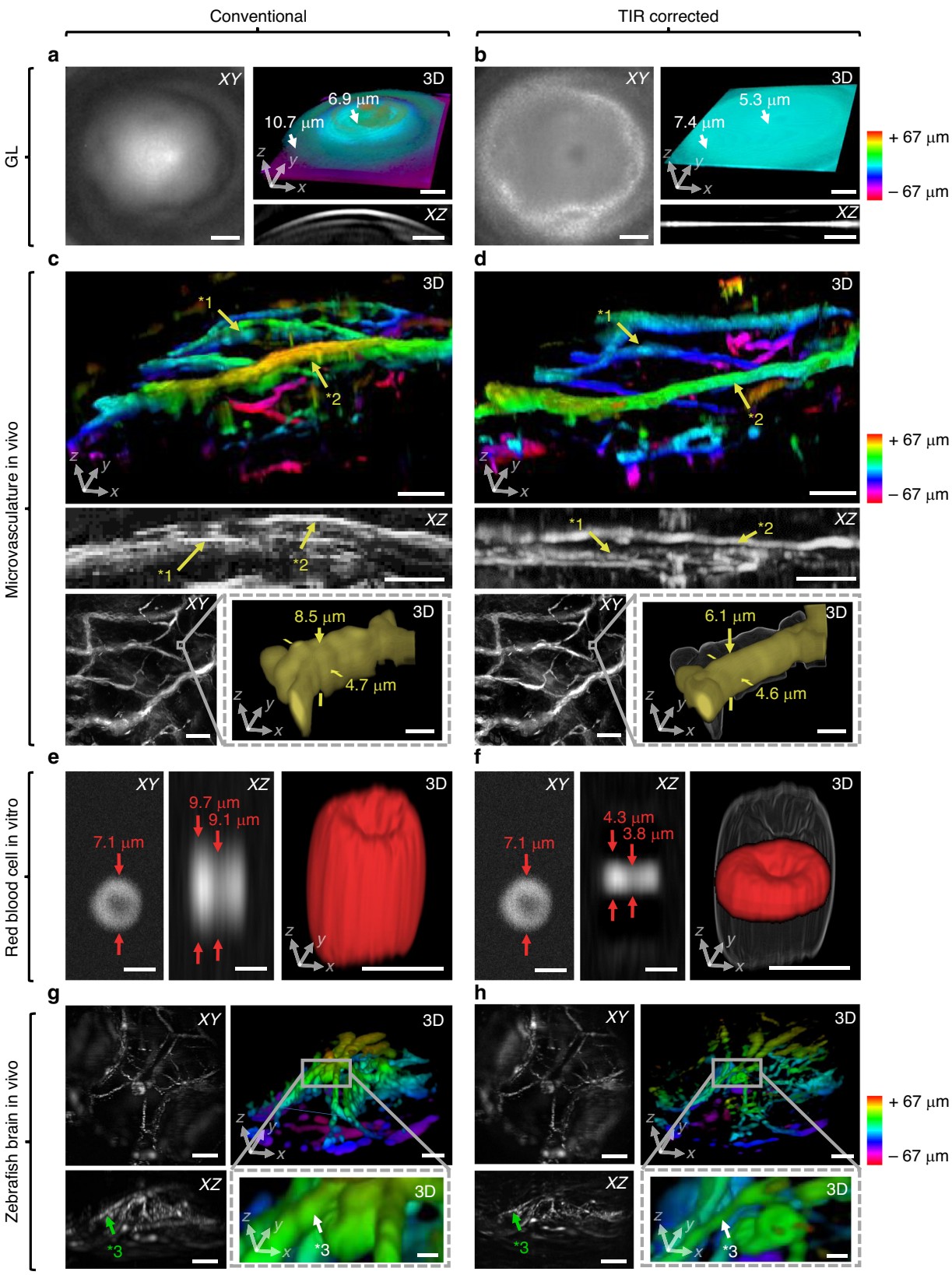

resolution features compared with conventional techniques. The new performance allows observations not before available to optoacoustic microscopy and the inspection of images with 3D accuracies not achieved so far in the field. This performance is based on a TIR-characterization method that scans diffraction-limited optical excitation in 3D across an ultra-thin GL to generate SOAPs, covering the entire sensitivity field of the ultrasound detector employed. In this way, the first complete optoacoustic microscopy TIR characterization has been also experimentally achieved.

**Fig. 7 Image improvement through TIR correction for experimental data of artificial and biological specimens.** Comparison between projections along the x, y, and z dimensions of 3D reconstructions of OR-optoacoustic microscopy images. Reconstructions were performed using conventional Hilbert transformation and TIR correction. Imaging of a gold layer (GL): **a** conventional reconstruction reveals strong field curvature, decreased axial resolution, especially at outer parts of the imaged FOV, and repetitive signal artifacts; **b** TIR correction eliminates field curvature and artifacts as well as significantly improves and equalizes the axial resolution throughout the FOV. Imaging of a mouse ear vasculature in vivo: **c** conventional reconstruction shows vessels blurred in z (see vessel *1) as well as strongly bend in z (see vessel *2); **d** TIR correction let vessel appear smooth (*1) and levelled (*2). Zoomed-in view of the indicated ROI, showing a microcapillary. Conventional reconstruction yields an elongated microcapillary in z, whereas TIR correction approximates a presumably circular microcapillary more accurate. Imaging of an isolated red blood cell (RBC) in vitro: **e** conventional reconstruction leads to an axially elongated RBC; **f** the TIR-corrected RBC appears flatter and smoother. Imaging of a zebrafish brain vasculature in vivo: **g** conventional reconstruction shows blurry vessels (see vessel *3); **h** TIR correction leads to a sharp vessel (*3). Scale bars, 100 μm (**a–d**, **g**) or 5 μm (**c**, **d** zoom-in, **e**, **f**); shadows in **d**, **f** zoom-ins are **c**, **e** zoom-ins, respectively.

While TIR determination and correction methods have been considered for optoacoustic macroscopy[28,41–45], no such method has been so far developed for optoacoustic microscopy. We show herein that acquiring and using the TIR for correcting microscopy images brings essential performance improvements in the image quality and ability to resolve structures along depth. The SOAP method is found superior to previous TIR-characterization approaches for optoacoustic microscopy, such as those involving pulse-echo recording, microsphere scanning, hydrophones, or numerical simulations, in terms of faster scanning, higher sensitivity, better reproducibility, more precise delineation of the 3D sensitivity field, and fewer positioning errors of the optoacoustic point source. Comparison of the experimental TIR characterization with the corresponding 3D simulation showed overall good agreement and revealed sources of interference due to reflections within the plano-concave glass lens used for acoustic focusing. Additional simulations of optoacoustic spherical sources of 1 μm size (solid or liquid; see Supplementary Fig. 10) indicate negligible differences in their associated signals, which further validates the usability of the GL as an emanation basis for optoacoustic point sources. The study further shows that an accurate TIR characterization should be performed experimentally in order to reveal the particular hardware performance of the system employed. For this reason, the SOAP method was designed as a seamless measurement that can be incorporated in the development and characterization of optoacoustic microscopes, enabling precise measurements of the spatial inhomogeneity of the acoustic sensitivity field as well as the electrical properties of the system.

Fed into an SMF algorithm for image correction, the obtained TIR function improved the SNR of optical-resolution optoacoustic microscopy by an order of magnitude and yielded at ~30% improvements in axial resolution, allowing for the first time more detailed observation of light-absorbing structures as a function of depth. Moreover, TIR correction reduced image distortions, position-dependent ToF differences, and temporal reflection artifacts. Since the experimental TIR data provides an accurate characterization of the system independently of the application, correction algorithms other than SMF, such as deconvolution, can make use of the TIR data. To even further refine and adapt the TIR characterization for the imaging of a specific specimen, a phantom mimicking the specimen could be easily positioned above the GL to account for acoustic deviation from the ideal coupling medium used herein, i.e., water. A similar approach could be attempted to address the optical deviations between biological specimens and the GL by redesigning the GL with an additional layer to incorporate optical characteristics of biological tissue. In the future, the TIR function could also be employed in model-based optoacoustic imaging to further improve image quality and overall performance and could serve as a characterization method for high-frequency ultrasound devices.

The SOAP method and subsequent 4D-TIR correction bring out new performance ability and we propose it as an essential treatment of optoacoustic microscopy data for improving the accuracy and fidelity of the images produced. Moreover, this strategy may well complement hybrid optical and optoacoustic microscopy systems by better matching the axial resolution achieved by the different modalities.

## Methods

**Theoretical TIR consideration.** The optoacoustic pressure wave generation and propagation can be generally expressed as[46]:

$$\frac{\partial^2}{\partial t^2}p(\mathbf{r}, t) - c^2\nabla^2 p(\mathbf{r}, t) = \Gamma\frac{\partial}{\partial t}H(\mathbf{r}, t), \quad (2a)$$

$$\forall H(\mathbf{r}, t) = \eta_h \cdot \mu_a(\mathbf{r}) \cdot \frac{\partial}{\partial t}\phi(\mathbf{r}, t), \quad (2b)$$

where $p$ denotes the local pressure, $c$ the speed of sound, $\Gamma$ the Grüneisen parameter, $H$ the heating function, $\eta_h$ the heat conversion efficiency (usually taken as 1 for simplicity), $\mu_a$ the optical absorption coefficient, and $\phi$ the light fluence distribution at the location $\mathbf{r}$ in space and at time $t$. For the optical-resolution optoacoustic microscopy case we can assume that the optical fluence is confined to a point in space. In addition, if the laser pulse is short enough for the heat and the stress confinement to meet, we can further assume that the distribution of $\phi$ in time is equivalent to a $\delta$ function. Then, Eq. (2a) is expressed as:

$$\frac{\partial^2}{\partial t^2}p(\mathbf{r}, t) - c^2\nabla^2 p(\mathbf{r}, t) = \Gamma\delta(\mathbf{r} - \mathbf{r}')\frac{\partial\delta(t)}{\partial t}. \quad (3)$$

The solution of Eq. (3) for each scanned position $\mathbf{r}'$ corresponds to the generated optical-resolution optoacoustic microscopy signal $p_0(\mathbf{r}, t)$ and can be expressed as[46]:

$$p_0(\mathbf{r}, t) = \frac{\Gamma}{4\pi c^2|\mathbf{r} - \mathbf{r}'|} \cdot \frac{\partial}{\partial t}\delta\left(t - \frac{|\mathbf{r} - \mathbf{r}'|}{c^2}\right). \quad (4)$$

Optical-resolution optoacoustic microscopy signals $p_{det}(\mathbf{r}, t)$ sensed by an ultrasound transducer are a convolution process of an ultrasonic wave generated in the excited medium and the overall properties of the transducer[47]. In case of high-speed laser scanning optical-resolution optoacoustic microscopy within the sensitivity field of the transducer, the signals originate from positions with axial and lateral offsets from the acoustic focus. As the acoustic waves enter into the focusing element of the transducer, typically a plano-concave glass lens, distortions and interferences lead to alterations of both the raw signals and consequently the generated images.

These alterations are captured in the total impulse response TIR(**r**, t) of the transducer, describing a convolution of the initial pressure $p_0(\mathbf{r}, t)$ with the spatial impulse response SIR(**r**, t), which depends on the sensing position **r**, and with the electric impulse response **EIR**(t), which is spatially invariant and represents the effects of the electrical acquisition chain:

$$p_{det}(\mathbf{r}, t) = p_0(\mathbf{r}, t)*\text{SIR}(\mathbf{r}, t)*\text{EIR}(t) = p_0(\mathbf{r}, t)*\text{TIR}(\mathbf{r}, t), \quad (5)$$

where * denotes the convolution. Accordingly, the spatial frequency response describes the corresponding convolution in Fourier space as $\text{SFR}(\mathbf{r}, f) = \mathcal{F}(\text{SIR}(\mathbf{r}, t))$. Finally, a precise determination of the TIR(**r**, t) at each voxel can lead to:

$$p_{corr}(\mathbf{r}, t) = p_{det}(\mathbf{r}, t)*\text{TIR}^{-1}(\mathbf{r}, t) \cong p_0(\mathbf{r}, t), \quad (6)$$

where $p_{corr}(\mathbf{r}, t)$ is retrieved as a TIR-corrected version of the detected pressure $p_{det}(\mathbf{r}, t)$ and, thus, is rectified for signal artifacts anticipated at this particular position in space.

**Experimental TIR characterization**. We employed a previously developed optical-resolution optoacoustic microscope, as comprehensively described in[16–18] and schematically depicted in Fig. 1a. The system is based on raster scanning a diffraction-limited focused optical excitation by a 532-nm laser emitting 1.4-ns pulses (SPOT-10-200-532, Elforlight, Daventry, UK). Optical-resolution optoacoustic microscopy signals are detected by a spherically-focused piezoelectric transducer with a central frequency of 50 MHz (Sonaxis, Besancon, France) positioned above the sample in transmission mode and coupled to the sample by a water droplet (Fig. 1b). The system allows for precise 3D positioning of the transducer and the sample using high-precision motorized stages (MLS203-2 and MZS500, Thorlabs, New Jersey, USA; M-683.2U4 and M.501.1DG, Physik Instrumente, Karlsruhe, Germany). The system optimizes optical focusing (Plan Apochromat 10×, Zeiss, Jena, Germany) by controlling the fine drive of the utilized inverted microscope stand (AxioObserver.D1, Zeiss) using a rotational stage (DT-34, Physik Instrumente). The system scans the optical excitation spot via a set of galvanometric mirrors (6215 H, Cambridge Technology, Bedford, USA) and is fully controlled in Matlab (Matlab 2014a, Mathworks, Natick, USA).

For comprehensive 4D-TIR determination, we scanned a focused optical excitation across an area of $630 \times 630 \ \mu m^2$ on a GL with a resolution of $100 \times 100$ pixels to generate a pattern of SOAPs as described below. The transducer was positioned axially from a positive defocus of 1.15 mm to a negative defocus of 1.15 mm with respect to the acoustic focal distance in steps of 100 nm, which allowed scanning of the GL to fill a 3D volume with discretely-generated SOAPs. The optical-resolution optoacoustic microscopy signals were recorded with a signal averaging of 500 at a sampling rate of 900 MS s$^{-1}$ (ADQ412, SP Devices, Linköping, Sweden). All pulses of the 50-kHz laser were recorded in a streaming-like acquisition, such that scanning each 2D plane of the 4D-TIR characterization required only 100 s.

**Gold layer**. As the 2D absorbing plane for TIR characterization, we fabricated a GL using electron beam assisted deposition under high vacuum below $10^{-7}$ mbar (Metal Evaporator L560, Leybold, Köln, Germany). First, a 4-nm titanium layer was deposited at 0.2 Å s$^{-1}$ on top of a glass cover slip 170 μm thick, which served as an adhesion layer. This metal thickness transmits ~13% of the laser power at 532 nm at normal incidence. Then, a 250-nm layer of gold was evaporated at 0.2 Å s$^{-1}$ in order to maximize the absorption of the laser light (~29% at 532 nm). We chose these relatively low evaporation rates to ensure high crystalline quality of the deposited metal layers. We calculated the reflectivity and transmission values using Fresnel's equations and material constants from the literature[48,49]. We employed gold slugs with a purity of >99.99% (Agosi AG, Pforzheim, Germany) and a mixture of titanium slugs with purity of >99.995% (HMW Hauner, Röttenbach, Germany) and >99.98% (Material Research SA). We selected a layer thickness of 250 nm for the 2D absorbing plane as a good and highly controllable compromise to ensure robust and consistent properties of the absorbing layer, to constitute an optoacoustic point source when illuminating with an optical focus, and to provide good acoustic transmission of expected weak signals: first, assuming a thermal conductivity of ~300 W m$^{-1}$ K$^{-1}$ for gold in comparison to ~0.6 W m$^{-1}$ K$^{-1}$ for water and 0.9 W m$^{-1}$ K$^{-1}$ for borosilicate glass D 263 ®M, we hypothesized that the thicker the GL, the more durable and resistant the sample. Second, the penetration depth of the equipped optical excitation into gold is ~40 nm, which therefore constitutes an axial point source independent on the GL thickness. Moreover, to ensure best possible acoustic signal propagation, we opted for a thin GL, whereas water has an acoustic impedance of $1.5 \times 10^6$ kg m$^{-2}$ s$^{-1}$ and an acoustic attenuation coefficient at 0.0022 db cm$^{-1}$ MHz, gold exhibits values of $62.6 \times 10^6$ kg m$^{-2}$ s$^{-1}$ and 1.64 dB cm$^{-1}$ MHz. For best possible transmission of the acoustic waves from their origin, i.e. the lower side of the GL, to the water-coupled transducer, i.e. above the GL, a thin GL is beneficial. Based on these assumptions and previous experiments, we empirically found 250 nm as a suitable layer thickness.

**Spatially distributed optoacoustic point sources**. We directed a diffraction-limited, optically-focused beam onto the ultra-thin GL, leading to the excitation of a 3D volume with dimensions of ~600 nm laterally (optical diffraction limit) and ~40 nm axially (penetration of 532 nm light into gold; also known as the skin depth). The resulting optoacoustic signals originate from volumes much smaller than the acoustic resolution of the equipped ultrasound transducer (Fig. 1b). Therefore, the GL acts effectively as an infinite plane in which optoacoustic point sources can be generated at desired locations. These generated so-called SOAPs showed similar properties as real point sources such as microspheres of same size. We could precisely control the positions of the SOAPs to ensure that they evenly covered the entire 3D volume to be imaged for TIR characterization as described above. For testing the suitability of the GL to generate SOAPs, we imaged a blacked 1-μm bead (Polybead® Black Dyed Microspheres 1.00 μm, Polysciences, Inc, Warrington, USA) with identical imaging parameters.

**Simulation-based TIR characterization**. In order to explore how optoacoustic signals are altered in a spatially dependent manner, we simulated the experimental TIR characterization. To enable comparable results, we designed a 3D simulation grid of $800 \times 800 \times 2300 \ \mu m^3$ in the k-wave extension of Matlab with a geometry identical to that of our performed experiment (Fig. 4a). We shaped a simulated

transducer according to the technical drawing of the ultrasound transducer equipped in the microscope setup. Analogous to the active element in the real transducer, the simulated transducer contained a disc of 3-mm diameter as the detection element. Simulated acoustic waves arriving at this disc were integrated and recorded. Furthermore, a cylindrical volume with acoustic properties of glass (speed of sound, 5200 m s$^{-1}$; density, 2500 kg m$^{-3}$) was defined with a base adjacent to the simulated sensor in order to represent the acoustic focusing lens of the transducer. Based on the manufacturer-provided information on the transducer's data sheet, the simulated acoustic focusing lens had a diameter of 5 mm and thickness of 4 mm, and it comprised a conical bevel with an angle of 45° over the lower 1.3 mm (Fig. 4a). At the end of cylinder opposite to the end adjacent to the sensors, the cylinder contained an inwardly bent concave surface with a radius of 1.1 mm and a depth 0.5 mm, leading to a rim of ~0.3 mm between the concave surface and the bevel (Fig. 4a). Apart from the simulated glass lens of the transducer, the simulation grid was defined with acoustic properties of water (speed of sound, 1500 m s$^{-1}$; density, 1000 kg m$^{-3}$), which was the acoustic coupling medium used in the experiment.

Next, we positioned acoustic point sources of 10 Pa initial pressure successively throughout a vertically-oriented 2D plane of $400 \times 2300 \ \mu m^2$ adjacent to the central axis of the simulated transducer (Fig. 4a). In this way, acoustic point sources were simulated at all experimentally measured lateral and axial offsets. Each simulation was carried out with a maximum supported frequency of 100 MHz and a dynamically-adjusted simulation duration to ensure that the acoustic waves entered the glass lens were reflected within it, and finally arrived at the simulated sensor. We further utilized the symmetric geometry of the simulation and projected the simulation results circularly around the central axis. In the end, we obtained a volume patterned with simulated point sources matching the experimental results.

**Mouse ear, zebrafish, and red blood cell imaging**. All animal procedures were approved by the Government of Upper Bavaria. An athymic nude Hsd *foxn1* mouse was anesthetized with 2% isoflurane. The ear of the mouse was placed at 500 μm positive defocus in the optical-resolution optoacoustic microscopy system using a custom-made 3D printed sample holder attached to the scanning unit of the inverted microscope. The mouse ear was raster scanned over an area of $630 \times 630 \ \mu m^2$ with a resolution of $400 \times 400$ pixels and a signal averaging of 20. The optical-resolution optoacoustic microscopy signal recording as well as processing were carried out analogously to the TIR measurements (Fig. 6). For TIR correction, each frequency-filtered A-scan was correlated with the corresponding frequency-filtered A-scan of the GL measurement at the same location, i.e., at the same axial and lateral offsets with respect to the transducer.

A larva of a zebrafish breed of maximum 6 days post fertilization (dpf) was paralyzed with 1%vol. of the muscle relaxant Flexeril (cyclobenzaprine hydrochloride, Sigma Aldrich) and embedded in low melting agar in a glass bottom petri dish prepared with egg water to prevent movements. The zebrafish scan was acquired with identical scan settings as the above-mentioned mouse ear.

A blood sample was acquired by finger pricking a healthy volunteer and performing a standard blood smear procedure on a 170-μm glass substrate. The measurement of the RBC was performed in an area of sparse occupation. The RBC scan was performed across a FOV of $32 \times 32 \ \mu m^2$ with a resolution of $400 \times 400$ pixels and a signal averaging of 20. The blood smear was positioned at the acoustic focus to take advantage of the higher sensitivity of the transducer when signal originate from the acoustic focus. For TIR correction, the GL was placed into the identical axial position of the acoustic focus and scanned accordingly.

**Reporting summary**. Further information on research design is available in the Nature Research Reporting Summary linked to this article.

## Data availability
The data that support the findings of this study are available from the corresponding author upon reasonable request.

## Code availability
The code that support the findings of this study is available from the corresponding author upon reasonable request.

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

## Acknowledgements

This work has been funded by the Deutsche Forschungsgemeinschaft (DFG) as part of the CRC 1123 (Z1). The authors would like to thank Sarah Glasl for helping with the mouse in vivo experiments, Anja Stelzl for helping with the zebrafish in vivo experiments, Dominik Jüstel for helping to implement the regularized signal deconvolution, and A. Chapin Rodriguez as well as Robert J. Wilson for their extensive support in developing the paper.

## Author contributions

M.S., D.S., and V.N. conceived the idea of TIR correction for optoacoustic microscopy. M.S. and D.S. designed the experiments and contrived the approach of generating spatially distributed optoacoustic point sources using a GL. M.S. performed all experiments of characterizing the TIR and measuring reference samples, the mouse ear microvasculature and the zebrafish larva in vivo as well as the RBC in vitro, carried out the TIR simulation, and applied the TIR correction using an SMF as well as alternative approaches for comparison. J.A. and G.D. conceptualized the SMF for TIR correction in optoacoustics and helped in its adaption to optoacoustic microscopy. J.W. manufactured the GL. M.S. and V.N. wrote the paper, and all authors read and edited the paper.

## Competing interests

The authors declare no competing interests.
