## [Peer Review File · Nature Communications]

Reviewers' comments:

Reviewer #1 (Remarks to the Author):

The manuscript by Seeger et al. investigated approaches to filter the impulse response in optoacoustic microscopy (OAM). This is an important but challenging task, because the impulse response in a OAM contains responses from both optical and acoustic aspects of the system. If an universal algorithm could be developed, it will certainly benefit the optoacoustic research field. However, reviewer has three major concerns regarding the work:

(1) A key validation for any algorithm is the in vivo imaging results. However, the data shown in Fig. 7 exhibits limited improvement. For instance, the ear images in the conventional approach show more vessels than those in the TIR corrected images. Unless authors could clearly show the improvement in the TIR method, it would be hard to convince the readers to use this technique.

(2) Various methods have been proposed to improve the 3D imaging capability of OAM. These methods include scanning the optical focus in depth direction, using Bessel beam instead of Gaussian beams to improve the depth of focus, 3D virtual detector reconstruction, and 3D vessel extraction and enhancement. All these approaches provided clear improvement in 3D image quality. How is the proposed method compare with other software or hardware-orientated solutions?

(3) Authors claimed that TIR is independent of application. However, the tissue's optical and acoustic properties will affect the TIR quantification. How could the variations in optical and acoustic properties be accounted for?

Because of these concerns, the reviewer feels that the article cannot be published in the current form.

Reviewer #2 (Remarks to the Author):

In this paper, the authors present a novel technique to characterise the impulse response of a photoacoustic microscope (with optical resolution). It covers the effects of the excitation optical focus, the detection ultrasound focus, and the electronic detection bandwidth altogether. This method uses virtual point sources, generated using an optically focused excitation of a gold nanolayer.

This enables to measure the spatial dependence of the PSF, as opposed to the previously introduced techniques, which characterised the PSF at the acoustic detector focus.

Finally, the authors experimentally demonstrate that this characterisation can be used to enhance the image quality, in terms of signal-to-noise ratio as well as resolution.

The paper is well written and technically sound. However the superiority of the proposed technique over previous ones should be more carefully backed up, and the paper contains many overstated claims.

Overall, I don't think it is suitable for publication in Nature Communications, as it addresses a rather technical issue and is not intended for a broad audience. Moreover, the enhancement of the imaging performances (although unquestionable) as well as the novelty of the approach are not significant enough to justify publication in such a journal.

Below are a few questions and comments about the paper that should in any case be addressed prior to publication.

Major comments:

* It can be misleading to speak about time dependence of the PSF. The authors are referring to the entangled roles of z and t in the acoustic measurements. They should make it clearer that they are not referring to a PSF which would be varying in time.

* Overall, the claim regarding the superiority of the proposed technique (in SNR and axial resolution, as in lines 415-416) is not adequately backed up. The paper would benefit from a proper experimental comparison between the different point sources generation methods, as proposed for instance in ref. 33. Moreover, methods developed in ref. 33 already allow for full 3D characterisation of the impulse response, according to the last paragraph of the paper.

The authors do verify here that a better SNR is achieved with the gold nanolayer as compared to a black polystyrene microsphere. But this only arises from the different absorption properties of the microsphere and the gold layer. And better SNR could easily be obtained by taking more measurements and averaging, which is not a problem when characterising the response of a system, or simply by using gold-coated microspheres.

I agree though that the use of a uniform layer is a more convenient than a collection of beads to characterise the PSF in an entire volume, but this is a much more tempered claim.

* How does the presented approach differ from "scanning of an optical focus across an absorber plane", which is said then to "not generate a pattern of proper 3D optoacoustic point sources" (l.136-138)? I assume the authors are referring to the much larger axial thickness of the generated point source because weaker absorbing material was used in ref.33, but it should be clarified to avoid confusion.

* The shape of the signals either generated with the bead or with the locally illuminated gold nanolayer actually differ by some non-negligible amount. It would be better to compare the signals with similar SNR, before stating anything about the equivalence of the 2 measurements.

* Moreover, the effect of the acoustic impedance mismatch between water and the gold layer is not mentioned, as it was for instance in ref.33. It could very much affect the final reconstruction, so it should be discussed at the very least.

* The authors claim artefacts of the TIR impose the use of 2D amplitude projections (see for instance l.150-155 or l.417-419), and that the 3D volume reconstruction is only enabled after correction using their method. This is obviously overstated: the supplemental videos show that 3D volumes can be reconstructed using the conventional method (frequency filtering + envelope detection). Although we certainly observe an effect of the total TIR correction, it does not reveal any feature that could not be distinguished beforehand.

* What motivated the choice of spatial matched filtering instead of standard deconvolution, using for instance a Lucy-Richardson algorithm?

* Strictly speaking, the method presented in this paper does not fully characterise the optical focus effect, as the effective lateral size of the point source is set by the optical focus itself. This actually corresponds to equation (4) in methods section, where $TIR = SIR * EIR$, but contradicts l.113-119, in which the authors include the optical impulse response.

I agree that using smaller objects will introduce various problems, as stated l.133-135, but the description of the characterisation method should be precise on this point, and at least consistent throughout the paper.

* In Fig.1:

- I don't see how the two measurements presented in c and d could be compared and what information could be learned from this.
- Please provide computation method of impulse responses (insets) and SNR in the methods.

* In Fig.3:

- What are dashed lines in h?

- Authors should mention how the various signals have been normalised, most likely with regard to the maximum value of each data set. It would be interesting to have the same figure when normalising everything to the global maximum value of the signal (at the acoustic focus). It would also be interesting to have similar figures (in supplementary material) with signals in a few specific bandwidths instead of the maximum amplitude.

* Sentence at lines 290-291 is a tautology, and does not provide any explanation regarding the offset bulge in the sensitivity profile.

* What useful information does the simulation provide? It is indeed insightful to see that simulated TIR does not exactly fit the measured one, but one could also raise the concern about missing features or over-simplifications in the simulation.

The authors should also compare the reconstructed images after spatial matched filtering using either the experimentally measured TIR or the simulated one.

Minor comments:

* l.129: references 31 and 32 are inadequately used here, as they do not involve any form of illumination. It seems that the authors were also referring to 33 and 34 as in the following sentence.

* l.131-134: There is no microsphere-based technique proposed in ref.34.

* One could eventually question the relevance of defining a new acronym referring to a collection of point sources (SOAPs)...

It may be a matter of taste, but I don't see any added value in contributing to the already overwhelming quantity of such acronyms when it is not fundamentally useful.

* typo l.294: "...explaining the relevancE of the features..."

* Fig.1: It would be better to set the less noisy trace (yellow) above the noisier one (black) for clearer display.

* Fig.3: In h: "central lobe" is not readable

Reviewer #3 (Remarks to the Author):

The subject manuscript reports on the performance improvement of optical-resolution optoacoustic microscopy (OR-OAM) by measuring the four-dimensional total impulse response (TIR). A measured TIR contains temporal and spatial information of the imaging systems limitation and thus applicable for spatially-dependent TIR correction of raw optoacoustic signals with a spatial matched filter. Thereby, as expected, the image axial resolution and SNR can be significantly improved. The article is interesting for the community, technically properly displayed and well written. However, due to the lack of scientific novelty it does not fulfill the criteria for publication in high impact nature research journals. The idea using measured or simulated TIR and spatial matched filter for image correction is not new. In summary, the article shows impressive the transfer of this method to OR-OAM with resulting improvements. As recommendation: The article is well suited for journals like Photoacoustics, Biomedical Optics Express, Journal of Biomedical Optics,.....where the competition is moderate compared to Nature Communication.

General comments:

- What is the specialty of the usage of the 250nm thick gold nano layer as target for measuring the SOAPs? Why is it called "gold nano layer" and not just "gold layer" as common. It is implausible why it need to be gold with a thickness of 250nm. I suppose every homogeneous highly absorbing layer

would to a good job for that purpose. Right?

- Line 171-173: Why does the layer thickness correlate with ablation threshold and the acoustic transmission? Provide more detailed explanation.
- Presumably the gold layer was deposited on a glass substrate and not free-standing. What about acoustic reflections and reverberations inside the glass substrate that might influence the impulse response?
- How sensitive is the performance of the method to any modifications of the 4D-TIR caused by acoustic inhomogeneities of the sample composition (speed of sound variations, attenuation), temperature effects, alignment variations, et cetera... ? Is it necessary to measure for each experimental setting the 4D-TIR in advance?
- What are the theoretical limitations of the method? What can be achieved under ideal conditions?
- Provide a more rigorous comparison to 3D deconvolution approaches, iterative methods used for photoacoustic imaging to boost the resolution and SNR?

We would like to thank the reviewers for the constructive feedback. We appreciate the thoughtful and positive comments, which have certainly helped to improve the presentation and quality of our paper. We have updated our paper according to the suggestions and performed more experiments as requested by the reviewers.

Answers to the reviewers in blue

Modifications of the manuscript in orange

Reviewer #1 (Remarks to the Author):

The manuscript by Seeger et al. investigated approaches to filter the impulse response in optoacoustic microscopy (OAM). This is an important but challenging task, because the impulse response in a OAM contains responses from both optical and acoustic aspects of the system. If an universal algorithm could be developed, it will certainly benefit the optoacoustic research field. However, reviewer has three major concerns regarding the work:

1. A key validation for any algorithm is the *in vivo* imaging results. However, the data shown in Fig. 7 exhibits limited improvement. For instance, the ear images in the conventional approach show more vessels than those in the TIR corrected images. Unless authors could clearly show the improvement in the TIR method, it would be hard to convince the readers to use this technique.

We agree with the reviewer that the paper would benefit from a clearer demonstration of the impact of the proposed method on *in vivo* samples and would like to draw his/her attention to the added supplementary information (Fig. S3 & S4), as well as the modified version of Fig. 7, with which we believe better highlights the achieved improvement. In this modified figure, some additional biological specimens (zebrafish brain vasculature and single red blood cell) have been corrected with our proposed method.

We added to l. 407-410 *"(..) various reference and unlabeled biological specimens (see supplementary information; Fig. S1 - S4), including the GL, 18 μm blacked polystyrene sutures, a mouse ear vasculature in vivo, an isolated red blood cell in vitro, and a zebrafish brain vasculature in vivo"*, and l. 440-448 *"A similar comparison between conventional reconstruction and TIR-correction was performed for an isolated red blood cell (RBC) in vitro (see Fig. 7e-f), which was imaged at the acoustic focus. Whereas the conventional reconstruction revealed an axially elongated RBC, TIR-correction achieved a much better representation of a RBC in form of a donut-like shape. Furthermore, the increase in SNR (see supplementary information; Fig. S1 - S4) led to a smoother surface boundary. Finally, an unlabeled zebrafish brain vasculature in vivo (Fig. 7.-h) was imaged and similarly compared. TIR correction achieved much sharper vessels and a more realistic geometry of the overall brain vasculature."*

2. Various methods have been proposed to improve the 3D imaging capability of OAM. These methods include scanning the optical focus in depth direction, using Bessel beam instead of Gaussian beams to improve the depth of focus, 3D virtual detector reconstruction, and 3D

vessel extraction and enhancement. All these approaches provided clear improvement in 3D image quality. How is the proposed method compare with other software or hardware-orientated solutions?

We thank the reviewer for this valuable comment and would like to draw his/her attention to the added supplementary information (Fig. S1 – S5). There, a comprehensive comparison between different methods of image improvement is provided, including several types of impulse correction and PSF correction methods. For that, we added in l. 449-468 *“The TIR-SMF approach was further compared to alternative methods that correct with the focal impulse response, with the simulated TIR, or with a TIR measured using a slab of absorbing material (with or without time-of-flight (ToF) correction) (see supplementary information; Fig. S1 – S4). For reference samples (i.e. GL and 18 μm suture) as well as biological specimens (zebrafish brain vasculature and red blood cell), the TIR-SMF approach led to the most accurate depiction in 3D and the highest SNR. In addition, the TIR-SMF approach was compared to deconvolution approaches both on the signal level as well as on the image level (see supplementary information; Fig. S5). For the former, standard numerical filtering methods like Fourier deconvolution proved to be not flexible enough to deal with the given situation and produced extremely noisy images. Hence, we formulated the deconvolution as a regularized linear least squares problem $f^* := \arg \min_f \|f * IR - s\|_2^2 + \lambda \|f\|_2^2$, where the regularization parameter was chosen visually from a broad range of values. The optimization was numerically solved with 5000 LSQR iterations to ensure convergence. For the image level, we characterized the systems spatially dependent PSF using a 1 μm microsphere at seven positions of lateral offset (i.e. 0 - 300 μm) and used a parallel iterative deconvolution in 3D to apply a modified residual norm steepest descent (MRSND) algorithm with five iterations and spatially dependent 3D PSFs. The TIR-SMF approach was again found to be superior to these alternatives in spatial representation of the data, noise reduction, and axial resolution improvement.”*

We have refrained from modifying the optical system, as the proposed method should be applicable to all versions of optoacoustic microscopy, independent of the particular setup, and thus, in theory, improve the performance of all those systems.

3. Authors claimed that TIR is independent of application. However, the tissue's optical and acoustic properties will affect the TIR quantification. How could the variations in optical and acoustic properties be accounted for?

We thank the reviewer for pointing this out. We agree that the optical and acoustic properties of the investigated specimens are not identical to the ideal case used herein for TIR characterization (i.e. a pure gold layer positioned on a glass substrate and water used for acoustic coupling). However, to refine the TIR characterization, phantoms mimicking biological specimens could be used to account for such deviations. We added in l. 514-519: *“To even further refine and adapt the TIR characterization for the imaging of a specific specimen, a phantom mimicking the specimen could be easily positioned above the gold layer to account for acoustic deviation from the ideal coupling medium used herein, i.e. water. A similar approach could be attempted to address the optical deviations between biological specimens and the GL by redesigning the GL with an additional layer to incorporate optical characteristics of biological tissue.”*

Because of these concerns, the reviewer feels that the article cannot be published in the current form.

Reviewer #2 (Remarks to the Author):

In this paper, the authors present a novel technique to characterise the impulse response of a photoacoustic microscope (with optical resolution). It covers the effects of the excitation optical focus, the detection ultrasound focus, and the electronic detection bandwidth altogether. This method uses virtual point sources, generated using an optically focused excitation of a gold nanolayer. This enables to measure the spatial dependence of the PSF, as opposed to the previously introduced techniques, which characterised the PSF at the acoustic detector focus. Finally, the authors experimentally demonstrate that this characterisation can be used to enhance the image quality, in terms of signal-to-noise ratio as well as resolution. The paper is well written and technically sound. However the superiority of the proposed technique over previous ones should be more carefully backed up, and the paper contains many overstated claims. Overall, I don't think it is suitable for publication in Nature Communications, as it addresses a rather technical issue and is not intended for a broad audience. Moreover, the enhancement of the imaging performances (although unquestionable) as well as the novelty of the approach are not significant enough to justify publication in such a journal.

Below are a few questions and comments about the paper that should in any case be addressed prior to publication.

Major comments:

1. It can be misleading to speak about time dependence of the PSF. The authors are referring to the entangled roles of z and t in the acoustic measurements. They should make it clearer that they are not referring to a PSF which would be varying in time.

We thank the reviewer for pointing out possible misconceptions induced by the definition of the photoacoustic PSF. We therefore corrected the following passages:

- l. 31: *“However, the time-resolved and bi-polar nature of photoacoustic signals challenges the deconvolution process. In particular, correction algorithms must take into account the dependence of signals on space and their profile in time, i.e. the four-dimensional total impulse response (TIR) of the microscope.”*
- l. 95: *“**First**, photoacoustic signals exhibit a time profile that contain depth information whereby optical signals are time-independent and therefore represent a single spatial point in the imaging plane.”*
- l. 122: *“The convolution of these three contributions for all points in the volume of interest is termed the total impulse response (TIR) of the imaging system and is therefore a 4D array (x, y, z, t) containing time-resolved photoacoustic impulse responses at all positions throughout the volume.”*

2. Overall, the claim regarding the superiority of the proposed technique (in SNR and axial resolution, as in lines 415-416) is not adequately backed up. The paper would benefit from a proper experimental comparison between the different point sources generation methods, as proposed for instance in ref. 33. Moreover, methods developed in ref. 33 already allow for full 3D characterisation of the impulse response, according to the last paragraph of the paper. The authors do verify here that a better SNR is achieved with the gold nanolayer as compared to a black polystyrene microsphere. But this only arises from the different absorption properties of the microsphere and the gold layer. And better SNR could easily be obtained by taking more measurements and averaging, which is not a problem when characterising the response of a system, or simply by using gold-coated microspheres. I agree though that the use a uniform layer is a more convenient than a collection of beads to characterise the PSF in an entire volume, but this is a much more tempered claim.

We thank the reviewer for his/her suggestion to compare our proposed method to existing attempts to account for the OAM-TIR. We therefore compiled a comprehensive comparison with alternative approaches in the form of supplementary information (Fig. S1-4).

We compared different absorbers to determine the total impulse response and then applied spatial matched filtering. We also incorporated time-of-flight corrections when only a static (i.e. central) impulse response is utilized.

- The comparison includes two reference samples (gold layer and 18 μm polystyrene suture) as well as two biological samples (zebrafish brain vasculature and single red blood cell).
- The comparison covers the conventional way (i.e. Hilbert transform), time-of-flight correction, focal impulse response, focal impulse response with correcting for the time-of-flight differences, simulated total impulse response, total impulse response determined using an absorbing slab, and the gold-layer total impulse response.
- In all 4 specimens, the correction using the gold-layer total impulse response showed the best results in 3D representation, axial resolution improvement, and signal-to-noise ratio improvement, whereas non-spatially matching impulse responses led to artifacts (e.g. signal duplication) or noise induction.

Furthermore, we would like to draw the reviewer's attention to our statement in line 137-142, which specifies the difficulties in achieving similar precision of an experimentally determined TIR using a mechanical scan of microspheres. *"Mechanical scans of microspheres in 3D have been suggested for comprehensive determination of spatial dependence, but this method is quite challenging for microscopic characterization due to sensitivity issues, the small sizes of optical absorbers required, high-frequencies generated from small objects, and their lack of photo-stability after repeated exposure to high-intensity focused light"*.

3. How does the presented approach differs from "scanning of an optical focus across an absorber plane", which is said then to "not generate a pattern of proper 3D optoacoustic point sources" (l.136-138)? I assume the authors are referring to the much larger axial thickness of the generated point source because weaker absorbing material was used in ref.33, but it should be clarified to avoid confusion.

We thank the reviewer for the pointing out this confusing statement. We modified the mentioned sentence in l. 143-146: *“Likewise, scanning of an optical focus across an absorber plane of non-negligible thickness (i.e. larger than half the theoretical axial resolution; $\sim 3 \mu\text{m}$ at signals of max 110 MHz; e.g. a slab of light absorbing material) would not generate a pattern of proper 3D optoacoustic point sources.”*

4. The shape of the signals either generated with the bead or with the locally illuminated gold nanolayer actually differ by some non-negligible amount. It would be better to compare the signals with similar SNR, before stating anything about the equivalence of the 2 measurements.

We kindly disagree with the reviewer, as this comparison attempts to show the superiority of the gold layer over a microsphere. Due to that, all measurement parameters were kept equal to emphasize that an identical scan (i.e. pulse energy, lateral and axial position, number of averages, sampling rate, frequency filtering, and projection) leads to a significantly higher SNR. Furthermore, for optoacoustic microscopy, the characteristics of peak-to-peak amplitude, ratio, duration of the A-line, the bandwidth, and central frequency are essential for correction approaches on the signal level. As shown in Fig. 1 e and f, the scans of the microsphere and the gold layer resemble each other in these characteristics.

We further would like to emphasize that we do not claim equivalence, rather according to l. 227 merely a *“Good agreement (...)”*.

5. Moreover, the effect of the acoustic impedance mismatch between water and the gold layer is not mentioned, as it was for instance in ref.33. It could very much affect the final reconstruction, so it should be discussed at the very least.

In the figure below, the reviewer can find the simulation of the signal generated by a $1 \mu\text{m}$ solid diameter sphere made of gold, submerged in water and a $1 \mu\text{m}$ diameter liquid sphere filled with blood, assuming an instantaneous pulse and a transducer frequency range from 10 MHz to 120 MHz. The difference is minimal and probably stays within the noise level. We further added to l.495-498: *“Additional simulations of optoacoustic spherical sources of $1 \mu\text{m}$ size (solid or liquid; data not shown) indicate negligible differences in their associated signals, which further validates the usability of the GL as an emanation basis for optoacoustic point sources”*

6. The authors claim artefacts of the TIR impose the use of 2D amplitude projections (see for instance l.150-155 or l.417-419), and that the 3D volume reconstruction is only enabled after correction using their method. This is obviously overstated: the supplemental videos show that 3D volumes can be reconstructed using the conventional method (frequency filtering + envelope detection).

Although we certainly observe an effect of the total TIR correction, it does not reveal any feature that could not be distinguished beforehand.

We thank the reviewer for commenting on our statement, which indeed required clarification. We changed the statement in l. 160-164 to *“As a result, optoacoustic microscopy images today are commonly rendered as frequency-filtered maximum amplitude projections (MAPs), which are typically displayed as top-view two-dimensional images and conceal complex three-dimensional features.”* ; and the statement in l. 474-476 to *“The new performance allows observations not before available to optoacoustic microscopy and the inspection of images with 3D accuracies not achieved so far in the field”*. Furthermore, we would like to draw the reviewer’s attention to the added supplementary information showing a detailed comparison between different correction methods including 3D representations. (see reviewer #1 point 1).

7. What motivated the choice of spatial matched filtering instead of standard deconvolution, using for instance a Lucy-Richardson algorithm?

We thank the reviewer for this valuable comment and would like to draw his/her attention to the added supplementary information. There, we compared different algorithms to correct for the systems total impulse response both on the signal level using the TIR (spatial matched filtering, LSQR) as well as on the image level using the spatial PSF (MRSND) (see reviewer #1 point 2).

8. Strictly speaking, the method presented in this paper does not fully characterise the optical focus effect, as the effective lateral size of the point source is set by the optical focus itself. This actually corresponds to equation (4) in methods section, where $TIR = SIR * EIR$, but contradicts l.113-119, in which the authors include the optical impulse response. I agree that using smaller

objects will introduce various problems, as stated l.133-135, but the description of the characterisation method should be precise on this point, and at least consistent throughout the paper.

We thank the reviewer for this valuable comment. Indeed, our proposed approach does not incorporate the lateral shape of the optical focus and mainly concerns the axial properties of OAM. Further, TIR correction does not affect the lateral sharpness of the images (or in the best case due to SNR improvements), because this is, as correctly pointed out by the reviewer, defined by the optical focus and, thus, can be corrected for subsequent to TIR correction via e.g. spatial 2D Wiener deconvolution in each image layer.

To clarify the definition of the OIR, we added in l. 123-129: *“The lateral shape of the PSF for optoacoustic microscopy is defined by the optical focus and, thus, is independent of acoustic properties. Accurate TIR correction would therefore mainly affect the SNR and the axial projection of the data, whereas the lateral representation could be subsequently corrected with a standard spatial Wiener deconvolution. In the context of TIR correction for optoacoustic microscopy, the OIR can thus be reduced to its axial component, which is entangled with the laser pulse duration.”*

We further would like to draw the reviewer’s attention to the added supplementary information comparing the herein proposed TIR correction on the signal level to a spatial deconvolution using the systems spatial PSF (see reviewer #1 point 2).

9. In Fig.1:

- I don't see how the two measurements presented in c and d could be compared and what information could be learned from this.

The two panels c and d show scans of either the microsphere and the gold layer.

For clarification, we added l.219-226 *“Scanning the GL at a given axial distance to the transducer consequently characterizes several orders of magnitude more spatial locations of the system’s TIR than an equivalent scan of a single microsphere. Hence, determining the entire TIR in 3D only requires axial displacement of the GL in combination with 2D lateral scans, which constitutes a significantly faster process than a mechanical scan of the microsphere. Furthermore, Fig. 1d illustrates, that a 2D scan of the assumingly homogenous GL appears inhomogeneous, which could indicate imperfections in the transducer.”*

The panels are supposed to demonstrate that a scan of a single microsphere characterizes the system’s impulse response only at one position (i.e. the location of the point source), whereas the scan of the gold layer achieves this for all scanned pixels. Consequently, scanning the gold layer can determine the impulse response at > 5 orders of magnitude more locations for a given axial distance between gold layer or transducer, which reduces the required measurement time significantly. Furthermore, the scan of the gold layer shows the inhomogeneity of its capturing, most probably induced by imperfections of the transducer, as the gold layer itself can be assumed as perfectly homogenous.

- Please provide computation method of impulse responses (insets) and SNR in the methods.

The impulse responses (inset of Fig 1e) are generated by time-wise integration of the recorded time-resolved A-scans; the associated frequency response is yielded as its Fourier transform. We added in l. 233 “, *yielded via time-wise integration [30]*” and in l. 234 “*as the impulse response’s Fourier transform*”. The SNR for this experiment was calculated as the maximum amplitude of the signal (record window 2.9 – 3.2 μ s after the trigger signal) over the maximum amplitude over the noise (identical record window with blocked laser beam), as usually performed in the field. We added in l. 230 “, *via comparing the maximum amplitude of the signal recorded in the time window from 2.9 - 3.2 μ s after the trigger signal to an identical recording with a blocked laser beam*”

10. In Fig.3:

- - What are dashed lines in h?

The dashed lines were inserted to highlight the boundary between the central and the side lobes of the sensitivity field. However, we agree that the dashed lines could potentially be misinterpreted and have therefore removed them.

- - Authors should mention how the various signals have been normalised, most likely with regard to the maximum value of each data set. It would be interesting to have the same figure when normalising everything to the global maximum value of the signal (at the acoustic focus). It would also be interesting to have similar figures (in supplementary material) with signals in a few specific bandwidths instead of the maximum amplitude.

We thank the reviewer for this comment and added in l. 306 the expression “*each individually normalized*”. We further added non-normalized depictions of the 3D sensitivity field in the supplementary information for the entire frequency range (5-110 MHz) as well as low (5-40 MHz), mid (40-75 MHz), and high (75-110 MHz) bands. We further added l. 311-315 “*Generating the sensitivity field for certain frequency bands reveals a strong spatial dependence of the frequency coverage (see supplementary information; Fig. S6). The spatial side lobes are mostly carrying low frequencies, the bulge seems to be induced by middle frequencies, and the acoustic focus is dominated by high frequencies.*”

11. Sentence at lines 290-291 is a tautology, and does not provide any explanation regarding the offset bulge in the sensitivity profile.

We do not agree with the reviewer, as l. 309-311 states that we assume the knob-like “*bulge may be explained by the overlap of the spatial side lobes also visible in the side view of the central plane.*” We further added some frequency-dependent representations of the sensitivity field in the supplementary information and provide more insights into the characteristics of the

bulge. We added in l. 311-315 *“Generating the sensitivity field for certain frequency bands reveals a strong spatial dependence of the frequency coverage (see supplementary information; Fig. S6). The spatial side lobes are mostly carrying low frequencies, the bulge seems to be induced by middle frequencies, and the acoustic focus is dominated by high frequencies”*

12. What useful information does the simulation provide? It is indeed insightful to see that simulated TIR does not exactly fit the measured one, but one could also raise the concern about missing features or over-simplifications in the simulation. The authors should also compare the reconstructed images after spatial matched filtering using either the experimentally measured TIR or the simulated one.

We thank the reviewer for this comment and would like to draw his/her attention to the added supplementary information (Fig. S1-4). as well as our answers to reviewer #2 point 2.

Minor comments:

13. l.129: references 31 and 32 are inadequately used here, as they do not involve any form of illumination. It seems that the authors were also referring to 33 and 34 as in the following sentence.

We thank the reviewer for his/her thorough reading and corrected the wrongly placed references.

14. l.131-134: There is no microsphere-based technique proposed in ref.34.

We thank the reviewer for his/her thorough reading and corrected the wrongly placed reference.

15. One could eventually question the relevance of defining a new acronym referring to a collection of point sources (SOAPs)... It may be a matter of taste, but I don't see any added value in contributing to the already overwhelming quantity of such acronyms when it is not fundamentally useful.

We generally agree with the reviewer with respect to the merits of refraining from avoidable definitions of acronyms. However, as the generation of spatially-distributed optoacoustic point sources (SOAPs) and their optoacoustic properties are of essential importance for the present study and the acronym is used 30 times throughout the manuscript, we evaluate an acronym not only useful but also beneficial for straight forward grasping the versatility of the SOAPs.

16. typo l.294: "...explaining the relevanCE of the features..."

We thank the reviewer for his/her attention and corrected the typing error.

17. Fig.1: It would be better to set the less noisy trace (yellow) above the noisier one (black) for clearer display.

We thank the reviewer for the valuable suggestion and corrected the Fig. 1 e and f accordingly.

18. Fig.3: In h: "central lobe" is not readable

We thank the reviewer for his/her attention and changed to position of the label 'central lobe' accordingly.

Reviewer #3 (Remarks to the Author):

The subject manuscript reports on the performance improvement of optical-resolution optoacoustic microscopy (OR-OAM) by measuring the four-dimensional total impulse response (TIR). A measured TIR contains temporal and spatial information of the imaging systems limitation and thus applicable for spatially-dependent TIR correction of raw optoacoustic signals with a spatial matched filter. Thereby, as expected, the image axial resolution and SNR can be significantly improved. The article is interesting for the community, technically properly displayed and well written. However, due to the lack of scientific novelty it does not fulfill the criteria for publication in high impact nature research journals. The idea using measured or simulated TIR and spatial matched filter for image correction is not new. In summary, the article shows impressive the transfer of this method to OR-OAM with resulting improvements. As recommendation: The article is well suited for journals like Photoacoustics, Biomedical Optics Express, Journal of Biomedical Optics,.....where the competition is moderate compared to Nature Communication.

General comments:

1. What is the specialty of the usage of the 250nm thick gold nano layer as target for measuring the SOAPs? Why is it called "gold nano layer" and not just "gold layer" as common. It is implausible why it need to be gold with a thickness of 250nm. I suppose every homogeneous highly absorbing layer would to a good job for that purpose. Right?

We agree with the reviewer on simply naming it "gold layer". We changed the manuscript and the figures accordingly.

Indeed, other highly absorbing and spatially homogenous layers of material would perform equally good if not better. However, the quality and homogeneity of the layer, which is required to be two times thinner as the expected acoustic resolution ($< 3 \mu\text{m}$), have to be extremely high. Only such an absolute homogenous and flat absorbing layer with identical optical properties can ensure that spatially-distributed point sources generated by scanning an optical focus across the layer are as identical as possible. This is also stated in the manuscript in l. 172-175 "*For this*

reason, we selected to focus an optical beam onto an ultra-thin absorbing layer. This process generates spatially-distributed optoacoustic point sources (SOAPs), that are defined laterally by the diffraction limit of the optical excitation and axially by the light penetration into the absorbing layer”.

In order to clarify this point in the manuscript, we further added in l. 625-628 “We selected a layer thickness of 250 nm for the 2D absorbing plane as a good and highly controllable compromise to ensure robust and consistent properties of the absorbing layer, to constitute an optoacoustic point source when illuminating with an optical focus, and to provide good acoustic transmission of expected weak signals”.

2. Line 171-173: Why does the layer thickness correlate with ablation threshold and the acoustic transmission? Provide more detailed explanation.

We thank the reviewer for pointing out the relevance of the layer thickness, which helped us improve the quality of our paper. We added in l. 625-640: “We selected a layer thickness of 250 nm for the 2D absorbing plane as a good and highly controllable compromise to ensure robust and consistent properties of the absorbing layer, to constitute an optoacoustic point source when illuminating with an optical focus, and to provide good acoustic transmission of expected small signals: First, assuming a thermal conductivity of $\sim 300 \text{ W/m}\cdot\text{K}$ for gold in comparison to $\sim 0.6 \text{ W/m}\cdot\text{K}$ for water and $0.9 \text{ W/m}\cdot\text{K}$ for borosilicate glass D 263 [®]M, we hypothesized that the thicker the gold layer, the more durable and resistant the sample. Second, the penetration depth of the equipped optical excitation into gold is $\sim 40 \text{ nm}$, which therefore constitutes an axial point source independent on the gold layer thickness. Moreover, to ensure best possible acoustic signal propagation, we opted for a thin gold layer. Whereas water has an acoustic impedance of $1.5 \cdot 10^6 \text{ kg/m}^2\text{s}$ and an acoustic attenuation coefficient at 0.0022 dB/cm/MHz , gold exhibits values of $62.6 \cdot 10^6 \text{ kg/m}^2\text{s}$ and 1.64 dB/cm/MHz . For best possible transmission of the acoustic waves from their origin, i.e. the lower side of the gold layer, to the water-coupled transducer, i.e. above the gold layer, a thin gold layer is beneficial. Based on these assumptions and previous experiments, we empirically found 250 nm as a suitable layer thickness.”

3. Presumably the gold layer was deposited on a glass substrate and not free-standing. What about acoustic reflections and reverberations inside the glass substrate that might influence the impulse response?

We agree with the reviewer that some acoustic effects due to the multilayered composition of our gold plate arrangement (the arrangement was described in the methods section) might ruin the point like nature of the signal. In order to quantify such an effect, we compared the signal generated by our gold layer (plus titanium, plus glass; see methods), to the signal generated by a single microsphere (diameter = $1 \mu\text{m}$) embedded in agar (this was shown in results). Both signals were practically identical.

We further added in l. 495-498 “Additional simulations of optoacoustic spherical sources of $1 \mu\text{m}$ size (solid or liquid; data not shown) indicate negligible differences in their associated signals,

which further validates the usability of the GL as an emanation basis for optoacoustic point sources". See reviewer #2 point 5.

4. How sensitive is the performance of the method to any modifications of the 4D-TIR caused by acoustic inhomogeneities of the sample composition (speed of sound variations, attenuation), temperature effects, alignment variations, et cetera... ? Is it necessary to measure for each experimental setting the 4D-TIR in advance?

We thank the reviewer for this valuable comment and would like to refer to our response to reviewer #1 point 3.

5. What are the theoretical limitations of the method? What can be achieved under ideal conditions?

We thank the reviewer for this interesting idea. However, we would like to refrain speculating on the theoretical limitations of applying matched filters using the spatially dependent impulse response for optoacoustic microscopy. We think a comprehensive derivation would not be meaningful, especially in the scope of the here presented more general and practical attempt of improving optoacoustic microscopy, since various approximations/assumptions/simplifications of the matched filter theory do not hold for optoacoustics and, thus, would decrease the result's validity. To address the reviewers point, we simulated an ideal matched filter: Assuming a perfect N-shaped signal with added Gaussian noise and correcting with an analogously generated signal of

different noise. The unprocessed signals are characterized by an SNR of 43.4 dB and after matched filtering 93.1 dB. However, this simulation does neglect some critical aspects such as finite bandwidth of the transducer, non-Gaussian noise, or deviations from the ideal optoacoustic signal. These results are approximate at best, because of which we would like to refrain from including them in the manuscript or the supplementary information.

6. Provide a more rigorous comparison to 3D deconvolution approaches, iterative methods used for photoacoustic imaging to boost the resolution and SNR?

We thank the reviewer for this helpful comment and would like to draw his/her attention to the added supplementary information (SupFig. 1-5) as well as our answers to reviewer #2 points 2 and 7.

Reviewers' comments:

Reviewer #1 (Remarks to the Author):

The authors have made major changes to the experimental results and added comparison with existing techniques. Their modifications have addressed all my concerns. I have no further comments.

Reviewer #2 (Remarks to the Author):

Replies to specific points:

2. Figures S1, S2 and S3 are essential, the authors did a valuable effort carrying out this comprehensive comparison. The efficiency and superiority of their technique are undeniable, especially regarding SNR.

In Fig. S2, the signal repetition are still here, although one would expect such a correction would take care of this. It will actually be quite problematic when imaging vessels. Small deviations of the GL signal as compared to a real point source would be actually interesting to look at to understand artefacts in the final image (see replies to points 4 and 5 below).

4. I agree with the authors that their comparison exhibit the better performances of their technique regarding SNR, but I still emphasise that a proper comparison between to signal should be carried out as well with similar SNR, even though this would not be used in practical scenarios.

Moreover, in the third point of their reply to third reviewer, the authors actually state that the signals are "practically identical".

5. The simulation of signal generation from solid sphere of gold or liquid sphere of blood is helpful, but this does not fully answer my concerns. The layer structure is likely to matter, and the difference is already not so minimal as the authors state. The same concern was raised by reviewer 3, and minimum effort is required to perform the full simulation considering the gold layer and the 170um glass slab.

Overall, the image improvement when using the proposed correction is not striking enough to justify publication in Nature Communications, and technical issues would still require some investigations as mentioned above.

I encourage the authors to submit to a more technical journal, targeting a more restricted audience.

Reviewer #3 (Remarks to the Author):

After rigorous revision and supplementary information based on the reviewers comments the paper seems suited to be published in nature communications. Good job!

We appreciate the positive comments, acceptance, and agreement with the revised manuscript. We have updated our paper according to the remaining concerns of reviewer #2.

Initial review in green

Answers to the reviewers in blue

Modifications of the manuscript in orange (We kept the tracked changes of the initial review)

Reviewer #1 (Remarks to the Author):

The authors have made major changes to the experimental results and added comparison with existing techniques. Their modifications have addressed all my concerns. I have no further comments.

We thank the reviewer for accepting and agreeing to our changes and extensions of the manuscript. We are pleased to hear that our revised version addresses all raised concerns and would like to thank again the reviewer for the constructive feedback.

Reviewer #2 (Remarks to the Author):

Replies to specific points:

2. Figures S1, S2 and S3 are essential, the authors did a valuable effort carrying out this comprehensive comparison. The efficiency and superiority of their technique are undeniable, especially regarding SNR.

We thank the reviewer for the positive feedback on our effort to demonstrate the superiority of our proposed method.

In Fig. S2, the signal repetition are still here, although one would expect such a correction would take care of this. It will actually be quite problematic when imaging vessels. Small deviations of the GL signal as compared to a real point source would be actually interesting to look at to understand artefacts in the final image (see replies to points 4 and 5 below).

We thank the reviewer for pointing out “signal repetitions” in e.g. Fig. S2 (highlighted with red arrows). These signal repetitions are due to back-reflected ultrasound waves from the glass surface of the sample-holding petri dish. Hence, these signals are not “artefacts” or correlation side lobes induced by the spatial matched filtering with the spatially associated impulse response, but an actual signal detected by the transducer. This assumption is confirmed by observing the asymmetric distance in space (or time) of the original signal (highlighted with white arrows) and its back-reflection (highlighted with red arrows) in Fig. S2 panel a.2-g.2. To explain better this point, we added the following sentence in the caption of Fig. S2: *(..) induced by the ultrasound waves back-reflected from the sample-holding glass substrate (..)*

When SOAPS are generated directly on the GL, there are no acoustic back-reflections, as there is no spatial gap between the GL and the glass substrate. Therefore, only direct acoustic signals are recorded.

To rectify the signal repetitions induced by ultrasound waves back-reflecting from the sample-holding surface, one could apply time-windowing on the recorded signal. However, herein we opted to show the performance of the proposed TIR-SOAP-method when using the entire signal captured.

In regard to the influence of such signal repetitions when imaging vessels, we note that the artifacts that are not induced by acoustic reflections (highlighted with yellow arrows in Fig. S2) are minimized by image reconstruction using the TIR.

Finally, we would like to draw the reviewer's attention to our comments on point 5 regarding the influence of the layered structure of the GL for generating OA point sources.

4. I agree with the authors that their comparison exhibit the better performances of their technique regarding SNR, but I still emphasise that a proper comparison between to signal should be carried out as well with similar SNR, even though this would not be used in practical scenarios.

Moreover, in the third point of their reply to third reviewer, the authors actually state that the signals are "practically identical".

Reviewer's initial comment: The shape of the signals either generated with the bead or with the locally illuminated gold nanolayer actually differ by some non-negligible amount. It would be better to compare the signals with similar SNR, before stating anything about the equivalence of the 2 measurements.

Initial response: We kindly disagree with the reviewer, as this comparison attempts to show the superiority of the gold layer over a microsphere. Due to that, all measurement parameters were kept equal to emphasize that an identical scan (i.e. pulse energy, lateral and axial position, number of averages, sampling rate, frequency filtering, and projection) leads to a significantly higher SNR. Furthermore, for optoacoustic microscopy, the characteristics of peak-to-peak amplitude, ratio, duration of the A-line, the bandwidth, and central frequency are essential for correction approaches on the signal level. As shown in Fig. 1 e and f, the scans of the microsphere and the gold layer resemble each other in these characteristics.

We further would like to emphasize that we do not claim equivalence, rather according to I. 227 merely a "Good agreement (...)".

As explained in the initial review, the comparison between the GL and a microbead serving as possible point sources attempts to show that the GL leads to significantly better SNR when using the identical measurement parameters. To evaluate the similarity, we calculated the L2-Norm between the two specimens and updated the manuscript in l. 236-241: *We analysed the similarity between the signals recorded using the GL and the microsphere as possible point sources via the relative squared L²-error defined as $\frac{\|Sig_{GL} - Sig_{microsphere}\|_2^2}{\|Sig_{microsphere}\|_2^2}$ as it is invariant with respect to Fourier transform (Parseval's identity), which is relevant for the contained frequency content. Such analysis yielded a relative error of 0.28, whereas analogous analysis between the microsphere and a non-point source specimen (slab) yielded 0.90 (see supplementary information; Fig. S8).* The associated time-wise centered signals used for this analysis are

shown below as well as in the Supplementary Information as Fig. S8.

5. The simulation of signal generation from solid sphere of gold or liquid sphere of blood is helpful, but this does not fully answer my concerns. The layer structure is likely to matter, and the difference is already not so minimal as the authors state. The same concern was raised by reviewer 3, and minimum effort is required to perform the full simulation considering the gold layer and the 170um glass slab.

We thank the reviewer for the valuable comment and performed a simulation including a 170 μm glass slab beneath and directly attached to the simulated point sources. The simulation was performed analogous to the 3D TIR simulation (Fig. 4) at 6 positions within the simulated volume. Results are shown in the Supplementary Information as Fig. S7. We added further in the manuscript at l. 341-346: *We further performed a 3D simulation at 6 selected positions within the simulated volume that includes a 170 μm glass substrate to test the influence of the layered structure of the GL (see supplementary information; Fig. S7). Despite small additional signal artefacts that might be induced by acoustic reflections within the glass layer, no significant difference was observed. This simulation confirms the initial assumption of the GL serving as an emanation basis for appropriate OA point sources.*

Overall, the image improvement when using the proposed correction is not striking enough to justify publication in Nature Communications, and technical issues would still require some investigations as mentioned above.

I encourage the authors to submit to a more technical journal, targeting a more restricted audience.

Reviewer #3 (Remarks to the Author):

After rigorous revision and supplementary information based on the reviewers comments the paper seems suited to be published in nature communications. Good job!

We thank the reviewer for accepting and agreeing to our changes and extensions of the manuscript. We are pleased to hear that our revised version addresses all raised concerns and would like to thank again the reviewer for the constructive feedback.

REVIEWERS' COMMENTS:

Reviewer #2 (Remarks to the Author):

2. The authors have now made this point perfectly clear in their reply and most importantly in the paper.

I agree that there would be no such artefact when considering real thick biological sample.

4. The additional paragraph and Fig.S8 now provide an objective metric, and let the readers have their own appreciation of the similarity of the signals.

5. These simulations now fully address this issue, and again let the potential users decide whether they can afford this kind of signal distortion.

The authors have addressed all my concerns. Despite the substantial amount of work that they have been carrying out, it does not change my overall opinion about the significance of the paper and the relevance of a publication in a high impact and broad audience-targeted journal such as Nature Communications.